# An inhibitory circuit from central amygdala to zona incerta drives pain-related behaviors in mice

Sudhuman Singh[1], Torri D Wilson[1], Spring Valdivia[1], Barbara Benowitz[1], Sarah Chaudhry[1], Jun Ma[2], Anisha P Adke[1], Omar Soler-Cedeño[1], Daniela Velasquez[1], Mario A Penzo[2], Yarimar Carrasquillo[1,3]*

[1]National Center for Complementary and Integrative Health, Bethesda, United States; [2]National Institute of Mental Health, Bethesda, United States; [3]National Institutes on Drug Abuse, National Institutes of Health, Bethesda, United States

**Abstract** Central amygdala neurons expressing protein kinase C-delta (CeA-PKCδ) are sensitized following nerve injury and promote pain-related responses in mice. The neural circuits underlying modulation of pain-related behaviors by CeA-PKCδ neurons, however, remain unknown. In this study, we identified a neural circuit that originates in CeA-PKCδ neurons and terminates in the ventral region of the zona incerta (ZI), a subthalamic structure previously linked to pain processing. Behavioral experiments show that chemogenetic inhibition of GABAergic ZI neurons induced bilateral hypersensitivity in uninjured mice and contralateral hypersensitivity after nerve injury. In contrast, chemogenetic activation of GABAergic ZI neurons reversed nerve injury-induced hypersensitivity. Optogenetic manipulations of CeA-PKCδ axonal terminals in the ZI further showed that inhibition of this pathway reduces nerve injury-induced hypersensitivity whereas activation of the pathway produces hypersensitivity in the uninjured paws. Altogether, our results identify a novel nociceptive inhibitory efferent pathway from CeA-PKCδ neurons to the ZI that bidirectionally modulates pain-related behaviors in mice.

*For correspondence:
yarimar.carrasquillo@nih.gov

**Competing interest:** The authors declare that no competing interests exist.

## Editor's evaluation

This manuscript from Singh and colleagues investigates neural connections between the central amygdala and the zona incerta, two subcortical brain regions previously implicated in pain, and further describes the role of the zona incerta to preclinical pain-related behavior in mice. This study employed anatomical tracing, electrophysiology, optogenetics, chemogenetics, and behavioral assays in various pain modalities to link the zona incerta to pain modulation by providing new evidence for a direct inhibitory connection from the central amygdala to the zona incerta that could explain neuropathic pain hypersensitivity. This study is detailed anatomically, electrophysiologically, and behaviorally, and the inclusion of optogenetic studies has enhanced the conclusions. While there are still some confirmatory conclusions from prior work, the detail and execution of this study enhance the field.

## Introduction

Persistent pain resulting from lesions or diseases affecting the peripheral and central nervous systems can severely affect a person's life over time if left untreated (*Dworkin, 2002*; *Treede et al., 2008*). Understanding the neural circuits underlying pain processing and how they are recruited in a maladaptive manner following injury is crucial for the development of improved treatment options for

persistent pain. Several neuroimaging, pharmacological and electrophysiological studies in humans and animals demonstrate that the amygdala is a key locus in persistent pain processing (*Bernard and Besson, 1990*; *Bushnell et al., 2013*; *Carrasquillo and Gereau, 2007*; *Neugebauer et al., 2004*; *Zald, 2003*). A recent study further demonstrated that the central nucleus of amygdala (CeA) can both enhance and decrease pain-related behaviors in a cell-type-specific manner (*Wilson et al., 2019*). CeA neurons expressing protein kinase C-delta (CeA-PKCδ), for example, are sensitized by nerve injury and promote pain-related responses. In contrast, neurons expressing somatostatin are inhibited by nerve injury and promote decreases in pain-related behaviors. The circuit and cellular mechanisms responsible for bidirectional modulation of pain-related responses in the CeA, however, are still unclear.

In the present study, we began to address this question by characterizing the efferent projections from CeA-PKCδ neurons. Our cell-type-specific anatomical experiments identified the zona incerta (ZI) as one of the efferent targets of CeA-PKCδ neurons. The ZI is a subthalamic nucleus located ventrolateral to the medial lemniscus and dorsomedial to the substantia nigra (*Ricardo, 1981*). The ZI is comprised of heterogeneous groups of cells defined by the expression of molecular markers such as parvalbumin, tyrosine hydroxylase, somatostatin, calbindin, and glutamate (*Mitrofanis, 2005*). Previous studies using traditional anatomical tracing have shown that the ZI receives inputs from the CeA (*Reardon and Mitrofanis, 2000*). A recent study further demonstrated that somatostatin-expressing CeA neurons project to parvalbumin-expressing ZI neurons and contribute to conditioned fear memory (*Zhou et al., 2018*). The ZI has also been shown to modulate fear generalization (*Venkataraman et al., 2019*), binge eating (*Zhang and van den Pol, 2017*), defensive behaviors (*Chou et al., 2018*) and predatory hunting *Zhao et al., 2019* in rodents, highlighting the functional complexity of this subthalamic brain structure.

In rodent models of pain, changes in neural activity have been reported in the ZI (*Masri et al., 2009*) and behavioral studies further show that experimentally modulating the activity of ZI neurons alters behavioral hypersensitivity (*Hu et al., 2019*; *Moon et al., 2016*; *Petronilho et al., 2012*; *Wang et al., 2020*). Most of the literature suggests that the ZI is inhibited in the context of pain and that this inhibition drives behavioral hypersensitivity (*Hu et al., 2019*; *Masri et al., 2009*; *Moon et al., 2016*; *Moon and Park, 2017*). Consistent with this model, a recent study in humans showed that deep brain stimulation of zona incerta reduced experimental heat pain (*Lu et al., 2021*). Previous studies in rodents, however, reports the opposite – the ZI is activated in the context of pain and these increases in neuronal activity drive hypersensitivity (*Wang et al., 2020*). Together, these results suggest that modulation of pain in the ZI is complex and, most likely, cell-type and circuit-specific. Identifying the sources of excitation and inhibition of ZI neurons in the context of pain will be important to begin untangling the mechanisms underlying pain modulation in the ZI.

Based on our anatomical findings demonstrating projections from CeA-PKCδ neurons to the ZI, in combination with previous work showing that CeA-PKCδ neurons are GABAergic and display increases in activity following injury, we hypothesized that inhibitory inputs from CeA-PKCδ neurons are a source of pain-related inhibition in the ZI that results in increases in pain-related behaviors. In the present study, we tested this hypothesis using cell-type-specific anatomical tracing and optogenetically assisted circuit mapping along with chemogenetic or optogenetic manipulations coupled with behavioral assays to measure hypersensitivity in mice. Our combined results show that there is a functional inhibitory efferent pathway from CeA-PKCδ neurons to the ZI and that ZI-GABAergic neurons can bidirectionally modulate pain-related behaviors in mice. We further show that inhibitory inputs from CeA-PKCδ neurons to the ZI are necessary for ZI modulation of cuff-induced hypersensitivity and that activation of this pathway induced hypersensitivity in the absence of injury.

## Results
### Identification of CeA-PKCδ neuronal efferent targets

The neural pathways underlying modulation of pain-related behaviors by CeA-PKCδ neurons remains unknown. To begin to address this question, we mapped the anatomical localization of CeA-PKCδ terminals throughout the brain by stereotaxically injecting an adeno-associated virus (AAV) expressing the cre-dependent red fluorophore control mCherry into the CeA of PKCδ-cre mice (*Figure 1A*). We confirmed transduction of mCherry in CeA-PKCδ cells with immunostaining for PKCδ (*Figure 1B*) and subsequently analyzed the anatomical localization of mCherry-positive axonal terminals throughout

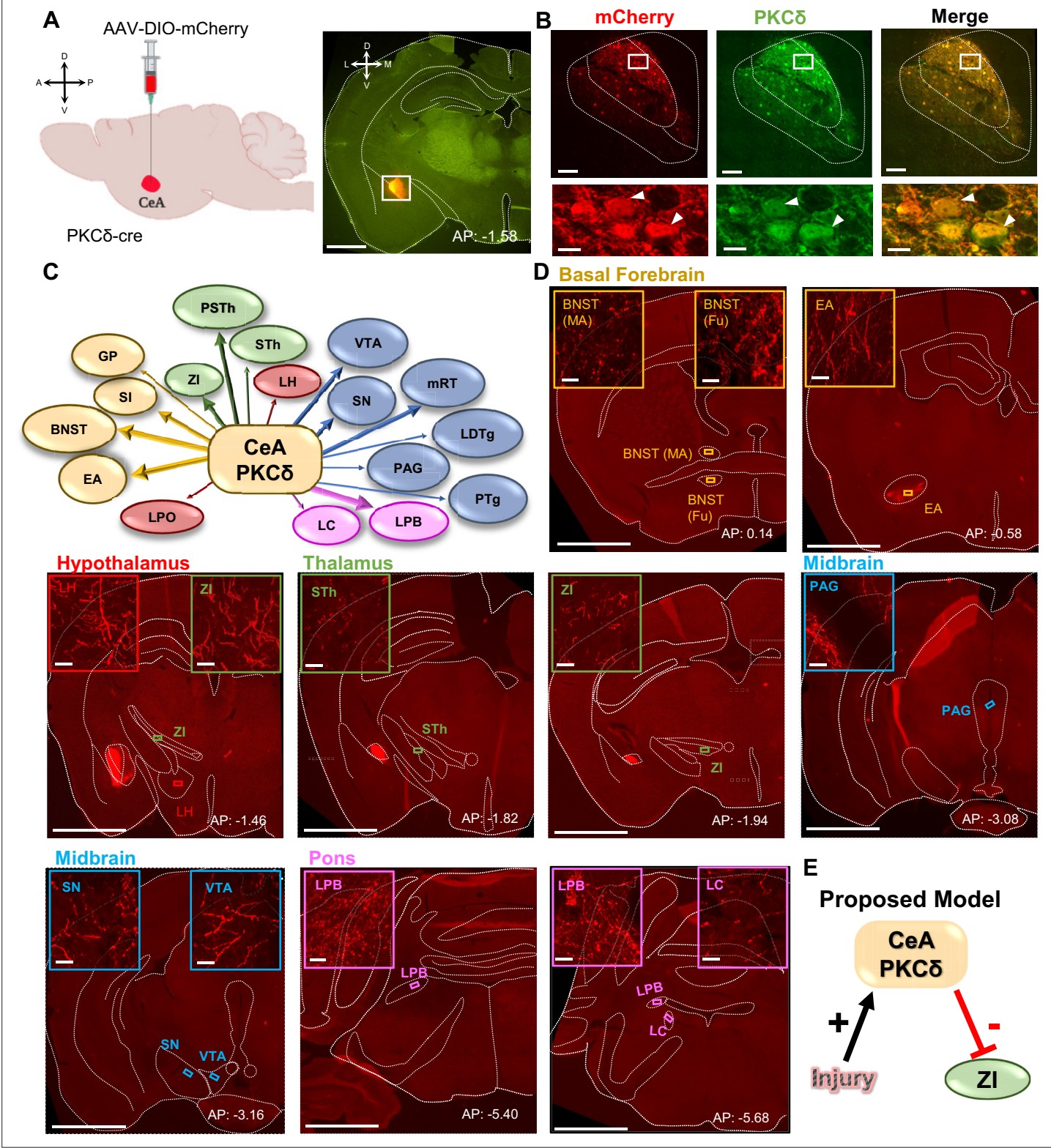

**Figure 1.** CeA-PKCδ neurons efferent targets. (**A**) Schematic of experimental approach. AAV-DIO-mCherry was unilaterally injected into the CeA of PKCδ-cre mice. A representative coronal brain slice of an injected mouse is shown on the right panel, with mCherry shown in red. Scale bar represents 1000 μm. (**B**) Representative high magnification images of the CeA in a coronal brain slice of an AAV-DIO-mCherry injected mouse. mCherry-transduced cells are shown in red and neurons immunostained for PKCδ in green. The merged image is shown on the right panel. Lower insets depict higher magnification images of the areas delineated by the white box in the upper images. White arrowheads highlight representative transduced cells that are

*Figure 1 continued on next page*

*Figure 1 continued*

also positive for PKCδ. Scale bars represent 100 μm for low magnification and 10 μm for high magnification images. (**C**) Summary diagram illustrating CeA-PKCδ neuron efferent projections within the brain. Forebrain regions are shown in yellow, hypothalamic structures in red, thalamus in green, midbrain in blue and pons in fuchsia. The thickness of the arrows depicts the density of labeling (sparse, moderate or dense). (**D**) Low magnification representative images of brain regions with axonal terminals from CeA-PKCδ cells. Insets in each image are high magnification images depicting axonal terminals within the regions delineated by the boxes in the respective low magnification images. Scales are 1000 μm for low magnification images and 20 μm for high magnification images. (**E**) Proposed model for pain-related inhibition of the ZI via injury-induced activation of CeA-PKCδ neurons. Abbreviations: bed nucleus of stria terminalis medial (BNST-MA); bed nucleus of stria terminalis fusiform nucleus (BNST-Fu); extended amygdala (EA); substantia innominata (SI); lateral preoptic area (LPO); globus pallidus (GP); lateral hypothalamus (LH); subthalamic nucleus (STh); zona incerta (ZI); parasubthalamic nucleus (PSTh); periaqueductal grey (PAG); substantia nigra (SN); ventral tegmental area (VTA); pedunculopontine tegmental nucleus (PTg); laterodorsal tegmental nucleus (LDTg); reticular formation (mRT); lateral parabrachial (LPB); locus coeruleus (LC). See *Figure 1—figure supplements 1 and 2*.

The online version of this article includes the following source data and figure supplement(s) for figure 1:

**Figure supplement 1.** Rostral-caudal distribution of viral injection sites within the CeA of mice used for anatomical experiments.

**Figure supplement 1—source data 1.** Source data for quantification of CeA neurons transduced with ChrimsonR-tdTomato or mCherry.

**Figure supplement 2.** Rostral-caudal distribution of CeA-PKCδ terminals in ZI.

**Figure supplement 2—source data 1.** Source data for quantification of CeA-PKCδ axonal terminal density within the ZI.

the brain. These analyses revealed CeA-PKCδ efferent projections in 18 brain regions, including the basal forebrain, striatum, thalamus, hypothalamus, midbrain, pons, and medulla (*Table 1*).

To validate these anatomical results, we stereotaxically injected two additional PKCδ-cre mice with an AAV expressing the cre-dependent red fluorescently tagged opsin ChrimsonR-tdTomato (*Figure 1—figure supplement 1A*). We selected ChrimsonR-tdTomato because it has been previously shown to reach terminals efficiently and functionally (*Li et al., 2022*). Similar to the results obtained in the mCherry experiments described above (*Figure 1B*), transduction of ChrimsonR-tdTomato was localized to CeA-PKCδ cells (*Figure 1—figure supplement 1B*). As summarized in *Table 1*, the anatomical distributions and terminal densities observed in mice injected with ChrimsonR-tdTomato were also comparable to those observed in mice injected with mCherry, showing that the identified regions are not dependent on the experimental approach used. In addition, mapping and quantification of transduced cells showed that transduction was selectively localized to the CeA, and that transduction efficiency was robust and comparable with both approaches (*Figure 1—figure supplement 1C-D*).

Semi-quantitative analysis, performed by visual examination of high magnification images, further revealed that axonal terminals in the output regions of CeA-PKCδ neurons have different terminal densities and organization patterns. Most of the brain regions identified had either few or moderate numbers of terminals, with only three regions, including the bed nucleus of stria terminalis, extended amygdala, and parabrachial nucleus, containing high densities of labeling (*Figure 1C–D* and *Table 1*).

As summarized in *Table 1*, moderate to dense labeling was consistently seen in the bed nucleus of stria terminalis, extended amygdala and CeA of all five brains analyzed. Dense labeling was also seen in the lateral parabrachial nucleus of three of the five brains analyzed, with sparse labeling observed in one brain and no labeling in the final brain. Sparse to moderate labeling was observed in the substantia innominata of all five brains; in the ZI, para-subthalamic nucleus, substantia nigra and reticular formation in four of five brains; and in the subthalamic nucleus and ventral tegmental area in three of the five brains analyzed. Lastly, sparse labeling was observed in the globus pallidus, lateral preoptic area and lateral hypothalamus of all five brains: in the locus coeruleus in four of five brains and in the pedunculopontine tegmental nucleus, laterodorsal tegmental nucleus, and periaqueductal grey in three of the five brains. Consistent with previous studies using traditional anterograde tracers in the CeA (*Aggleton, 2000*; *Barbier et al., 2017*; *Reardon and Mitrofanis, 2000*; *Shinonaga et al., 1992*; *Zhou et al., 2018*), no terminal labeling was observed in cortical regions of any of the five brains evaluated.

Mapping of the injection sites in all five brains shows that all injections were mostly restricted to the CeA (*Figure 1—figure supplement 1C*). Brains ET987, ET903, and ET832, however, had a more complete rostral-caudal coverage of the CeA than brain ET835 and the brain used in experiment 265945645 of the Mouse Brain Connectivity Atlas of the Allen Brain Institute, which mainly covered the posterior (but not anterior) portion of the CeA. Quantification of the number of transduced cells

**Table 1.** CeA-PKC $\delta$ neuronal efferent targets.

Semi-quantitative analysis of the density of axonal terminals in brain regions from 5 PKC $\delta$ -Cre mice stereotaxically injected with an adeno-associated virus expressing the cre-dependent gene (mCherry, ChrimsonR-tdTomato, EGFP) into the CeA. Rightmost column is from experiment 265945645 of the Mouse Brain Connectivity Atlas of the Allen Brain Institute (http://connectivity.brain-map.org/). - no expression;+sparse;++moderate;+++dense.

| Area | Abbreviations | ET832 mCherry | ET835 mCherry | ET 987 ChrimsonR | ET 903 ChrimsonR | Allen Brain Atlas EGFP |
|---|---|---|---|---|---|---|
| **Striatum and Basal Forebrain** | | | | | | |
| Bed nucleus of stria terminalis | BNST | +++ | +++ | +++ | +++ | +++ |
| Globus pallidus | GP | ++ | + | + | + | + |
| Extended amygdala | EA | +++ | +/++ | +++ | +++ | +++ |
| Central amygdala | CeA | ++ | +/++ | +++ | +++ | ++ |
| Substantia innominata | SI | ++ | +/++ | ++ | ++ | ++ |
| **Thalamus** | | | | | | |
| Subthalamic nucleus | STh | + | - | ++ | + | - |
| Zona Incerta | ZI | ++ | - | ++ | + | + |
| Para subthalamic nucleus | PSTh | +/++ | + | ++ | ++ | - |
| **Hypothalamus** | | | | | | |
| lateral preoptic area | LPO | +/++ | + | + | + | + |
| Lateral hypothalamus | LH | +/++ | + | + | + | + |
| **Midbrain** | | | | | | |
| Ventral tegmental area | VTA | +/++ | - | +/++ | + | - |
| Substantia nigra | SN | ++ | + | ++ | +/++ | - |
| Pedunculopontine tegmental nucleus | PTg | ++ | - | + | + | - |
| Laterodorsal tegmental nucleus | LDTg | - | + | + | + | - |
| Periaqueductal grey | PAG | - | + | +/++ | + | - |
| Reticular formation | mRT | +/++ | +/++ | +/++ | + | - |
| **Pons** | | | | | | |
| Lateral parabrachial | LPB | ++/+++ | + | +++ | +++ | - |
| Locus coeruleus | LC | ++ | + | + | + | - |

and analysis of the rostral-caudal distribution of labeling revealed that the number and distribution of neurons transduced with ChrimsonR-tdTomato and mCherry closely resembles the average number of tdTomato-positive cells quantified in sections from the PKCδcre mice crossed with the Ai9 reporter (*Figure 1—figure supplement 1D*). These results demonstrate that most of the CeA PKCδ-expressing cells were transduced with ChrimsonR-tdTomato or mCherry, allowing for an accurate evaluation of the terminals in these brains. Variability in the spread of viral transduction within the CeA might explain the differences observed in efferent projections between brains (*Table 1*), suggesting that projection-specific CeA-PKCδ neurons are topographically organized within the CeA.

## CeA-PKCδ neurons send inhibitory projections to the ZI

The ZI was among the brain regions identified as an efferent target of CeA-PKCδ neurons in our anatomical experiments (*Figure 1C–D*). These results were of interest because previous studies have shown that reduced activity in GABAergic ZI neurons correlates with pain-related behaviors (*Hu et al., 2019*; *Masri et al., 2009*; *Moon et al., 2016*; *Moon and Park, 2017*). Given that CeA-PKCδ neurons are GABAergic and are activated in the context of pain (*Wilson et al., 2019*), we hypothesized

that pain-related inhibition of the ZI is mediated by injury-induced activation of CeA-PKCδ neurons (*Figure 1E*).

The identification of the ZI as an efferent target of CeA-PKCδ neurons was somewhat surprising given that previous studies have reported that most of the ZI-projecting CeA neurons are somatostatin-positive, with only a small percentage of CeA-PKCδ neurons also projecting to the ZI (*Zhou et al., 2018*). To gain further insights into the anatomical projection from CeA-PKCδ neurons to the ZI, we qualitatively and quantitatively examined the anatomical distribution and densities of CeA-PKCδ axonal terminals within the ZI as a function of the rostral-caudal level. As illustrated in the representative images in *Figure 1—figure supplement 2B*, moderate CeA-PKCδ axonal terminal labeling was observed between rostral-caudal levels 1.46 mm and 2.70 mm posterior to bregma. Mapping of the distribution of CeA-PKCδ terminals within the ZI further showed that these axonal terminals were restricted to the middle and ventral sectors of the ZI in all brains analyzed (*Figure 1—figure supplement 2C*). Sparse terminal labeling was also seen throughout the whole ZI, independent of the mediolateral, dorsomedial, or rostral-caudal level. Lastly, quantification of terminal densities in areas with moderate terminal labeling showed a rostral-caudal gradient with greatest terminal densities observed at rostral-caudal level 1.46 posterior to bregma followed by decreasing densities up to rostral-caudal level 2.18 posterior to bregma, and a small increase in densities between rostral-caudal levels 2.46 and 2.70 posterior to bregma (*Figure 1—figure supplement 2D*). Together, these results indicate that CeA-PKCδ terminals are restricted to specific anatomical subregions within the ZI.

To validate these putative projections from CeA-PKCδ neurons to the ZI, we stereotaxically injected the retrograde tracer cholera toxin B (CTB) conjugated with Alexa Fluor 647 into the ZI of PKCδ-cre::Ai9 mice (*Figure 2A*). Evaluation of the anatomical distribution of CTB-positive CeA neurons showed that ZI-projecting CeA neurons are observed in all CeA subnuclei (capsular, lateral and medial) and are distributed along the rostral-caudal CeA (*Figure 2—figure supplement 1*). Higher densities of CTB-labeled neurons were observed in the middle sections of the CeA, between rostral-caudal levels 1.06 mm and 1.70 posterior to bregma (*Figure 2—figure supplement 1B*). Notably, CTB-647 was detected in CeA-PKCδ neurons (*Figure 2B*), validating the anatomical projection from CeA-PKCδ neurons to the ZI. Quantification of CeA neurons positive for both CTB-647 and PKCδ-td-Tomato further revealed that approximately 19% of CTB-positive CeA neurons are also positive for PKCδ-tdTomato. As illustrated in *Figure 2—figure supplement 1*, neurons positive for both CTB-647 and PKCδ-tdTomato were limited to the capsular and lateral subdivisions of the CeA, with higher densities observed in the middle/posterior regions of the CeA, between rostral-caudal levels 1.34 and 1.70 posterior to bregma. Together, the results from our anatomical experiments show that CeA-PKCδ neurons in the capsular and lateral CeA send inputs with moderate densities of terminals to the middle and ventral sectors of the ZI.

To characterize the functional connectivity between CeA neurons and ZI GABAergic neurons, we stereotaxically injected a virus expressing the excitatory opsin channelrhodopsin (ChR2-EYFP) into the CeA of VGAT-cre::Ai9 mice and performed patch-clamp recordings in acute brain slices containing the ZI (*Figure 2C*). The stable expression of ChR2-EYFP in the CeA is indicated by the presence of green, fluorescent signal in the peri-somatic region of VGAT-positive neurons in CeA. Consistent with our anatomical findings, tracing of the EYFP-labeled axonal terminals revealed moderate labeling of terminals in proximity of ZI neurons. Optogenetic stimulation of ChR2-expressing CeA terminals in the ZI with blue light further showed robust inhibitory post-synaptic currents in 60% of VGAT-positive cells (12 out of 20 cells; *Figure 2D*). Optically evoked postsynaptic responses occurred in the presence of TTX and 4-AP, demonstrating that the inputs from the CeA to the ZI are monosynaptic. Of note, recordings in VGAT-negative ZI neurons revealed that 53% (8 out of 15 cells) of these ZI neurons also display inhibitory post-synaptic currents in response to optical stimulation of ChR2-expressing CeA terminals. Quantification of the responses to optical paired stimulation further showed that paired pulse ratio is indistinguishable between VGAT-positive and VGAT-negative ZI neurons, demonstrating that CeA-synaptic inputs are comparable in VGAT-positive and VGAT-negative ZI neurons (*Figure 2D*).

To validate the inhibitory connectivity between the CeA and ZI, we recorded optically evoked inhibitory postsynaptic currents (oIPSC) of ZI neurons before and after bath application of the GABA$_A$ receptor antagonist bicuculline. As illustrated in the representative traces and graph in *Figure 2E*, the mean oIPSC amplitude in response to blue light stimulation (470 nm, 10 mW) was reduced after bath exchange to 10 µM bicuculline. Parallel recordings at a holding potential of –70 mV (near reversal

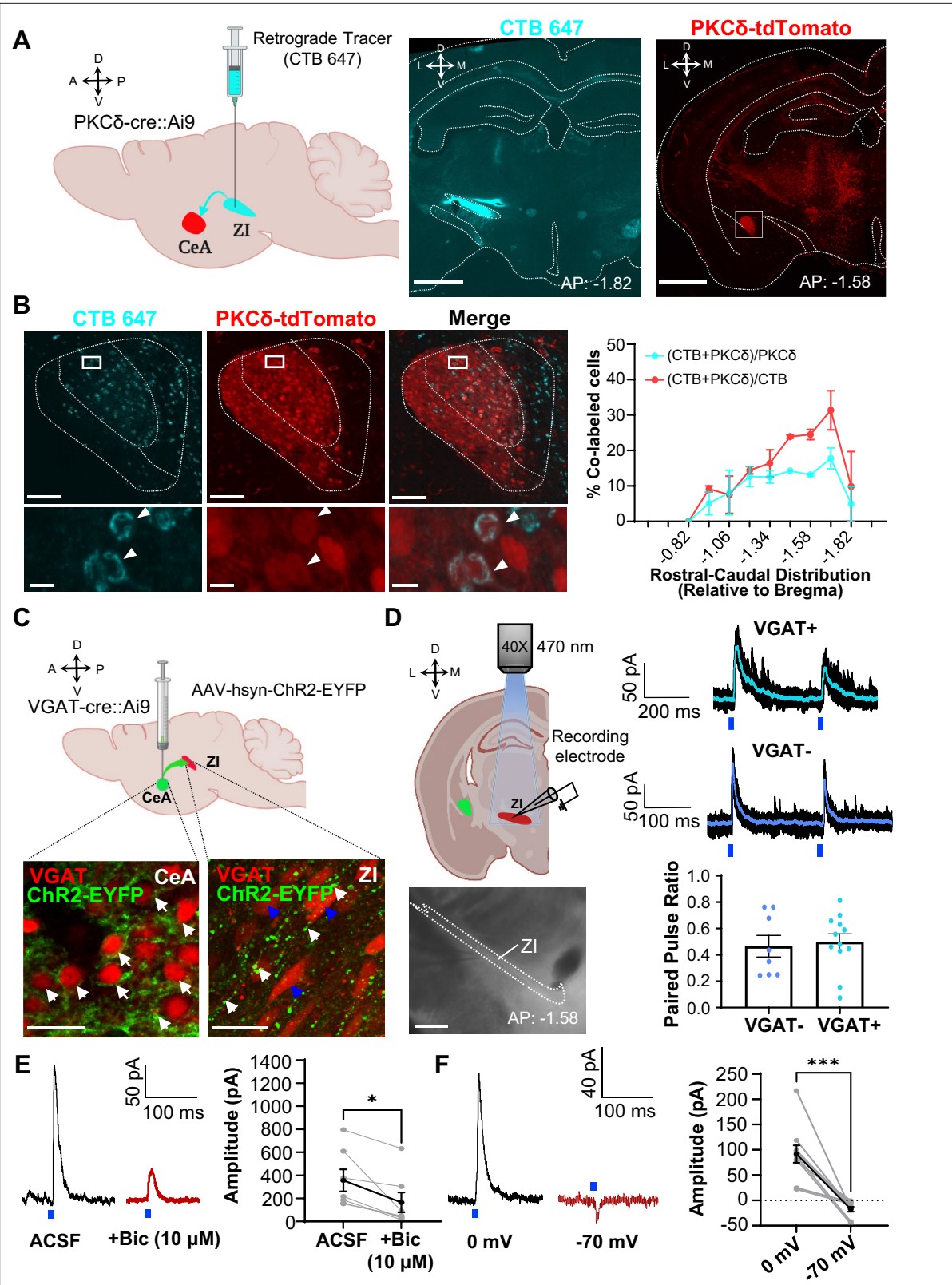

**Figure 2.** CeA-PKCδ inhibitory projections to the ZI (**A**) Schematic drawing of retrograde tracing experimental approach (left panel). Fluorescently tagged cholera toxin B (CTB-647) was injected into the ZI of a PKCδ-cre::Ai9 mouse brain. A representative coronal brain slice depicting the focal injection of CTB-647 (cyan) into the ZI is shown in the middle panel. A representative coronal brain slice containing the CeA is shown in the right panel. PKCδ-tdTomato cells are shown in red. The white square delineates the area magnified in panel **B**. Scale bar represents 1000 μm. (**B**) Representative

*Figure 2 continued on next page*

*Figure 2 continued*

high magnification images of the CeA in a PKCδ-cre::Ai9 mouse injected with CTB-647 into the ZI. CTB-positive cells are shown in cyan and PKCδ-tdTomato cells in red. The merged image is shown on the right. Lower insets show higher magnification images of the area delineated by the white squares in the top panel. Arrowheads highlight cells that are positive for CTB and PKCδ-tdTomato. Scale bars represent 100 µm (top panel) and 10 µm (bottom panel). The mean ± SEM percentage of PKCδ or CTB cells co-labeled for both PKCδ and CTB as a function of the rostral-caudal level is shown on the right (n=2 mice, 8 slices per mouse). (**C**) Schematics for the optogenetically assisted circuit mapping experiments. VGAT-cre::Ai9 and C57BL/6 J mice were stereotaxically injected with AAV-hsyn-hChR2-EYFP into the CeA. Lower left panel – perisomatic ChR2-EYFP (green) in VGAT-positive CeA neurons (red) are highlighted by white arrows. Lower right panel – CeA terminals (green; white arrows) in proximity to VGAT-positive ZI neurons (red; blue arrows). Scale bars are 20 µm. (**D**) Schematic diagram and differential contrast image of the ZI for ex-vivo whole-cell recordings in acute ZI brain slices is shown in the left panel. Scale bar is 500 µm. Top right panel - Representative traces showing responses of VGAT-positive (cyan) and VGAT-negative (blue) ZI neurons upon optical paired pulse stimulation of ChR2-expressing CeA terminals (0.5–10ms duration, 200ms inter-stimulus interval). Ten overlaid responses are shown in black and the averaged response in cyan or blue. Blue bars under the traces represent the timing and duration of blue light stimulation. The mean ± SEM paired pulse ratio is shown on the bottom right panel (n=8 VGAT-negative and 12 VGAT-positive cells). (**E**) Representative traces of optically evoked inhibitory postsynaptic currents (oIPSCs) of ZI neurons before and 3 min after bath exchange to ACSF containing 10 µM Bicuculline (Bic). The mean ± SEM of the oIPSC amplitude is shown on the right panel (n=7 cells collected from 7 mice; paired two-tailed t-test: t=3.361, df = 6, *p=0.0152 for ACSF vs Bicuculline). (**F**) Representative traces of light-evoked responses recorded at 0 mV and –70 mV. The mean ± SEM response amplitude is shown on the right panel (n=10 cells collected from 7 mice; paired two-tailed t-test: t=5.879, df = 9, ***p=0.0002 for 0 mV vs –70 mV). See *Figure 2—figure supplement 1*.

The online version of this article includes the following source data and figure supplement(s) for figure 2:

**Source data 1.** Source data for anatomical and electrophysiological validation of CeA-PKCd to ZI pathway.

**Figure supplement 1.** Anatomical validation of CeA-PKCd to ZI pathway using retrograde tracer approach.

**Figure supplement 1—source data 1.** Source data for quantification of CTB-647 and PKCδ-tdTomato positive neurons in the CeA.

potential for chloride) further showed either no response to blue light stimulation or a small inward current in neurons that showed robust light-evoked outward currents at holding potential of 0 mV (*Figure 2F*). Together, these findings confirm that the projection from the CeA to the ZI is inhibitory.

## Inhibition of ZI-GABAergic neurons is sufficient to induce bilateral hypersensitivity

Previous studies have shown that ZI-GABAergic neurons are inhibited in the context of pain (*Hu et al., 2019*; *Masri et al., 2009*; *Moon et al., 2016*; *Moon and Park, 2017*). To establish a causal link between reduced activity of ZI-GABAergic neurons and pain-related behaviors, we used a chemogenetic approach coupled with a battery of pain behavioral assays to measure tactile and thermal sensitivity in mice with the sciatic nerve cuff model of neuropathic pain or control mice that received a sham surgical procedure. To validate chemogenetic inhibition of ZI neurons, we performed whole-cell current-clamp recordings in acute ZI slices prepared from VGAT-cre mice stereotaxically injected into the ZI with an AAV encoding the cre-dependent inhibitory designer receptors exclusively activated by designer drugs (DREADD) hM4Di (*Figure 3A*). As illustrated in *Figure 3A*, bath application of the DREADD ligand clozapine N-oxide (CNO; 10 µM) significantly inhibited firing responses in hM4Di-transduced neurons, with no measurable effect observed in response to bath application of the saline vehicle control. Histological verification of injection sites at the end of the experiments further demonstrated that transduction of hM4Di was restricted to the ZI (*Figure 3B* and *Figure 3—figure supplement 1A*). The numbers and rostral-caudal distribution of transduced cells within the ZI of mice stereotaxically injected with hM4Di was comparable to the numbers and rostral-caudal distribution of VGAT-positive cells in the ZI of VGAT-cre::Ai9 mice, demonstrating robust transduction efficiency (*Figure 3C*).

We then measured the effects of selective chemogenetic inhibition of VGAT-positive ZI cells on pain-related responses to tactile and pressure stimulation of the hindpaws. This was done both before and after i.p. injection of either saline or CNO in both cuff and sham mice (*Figure 3D*). As expected, following cuff implantation on the sciatic nerve, tactile and pressure sensitivity was significantly lower in the hindpaw ipsilateral to cuff implantation compared to the contralateral hindpaw or either hindpaw in sham treated mice (*Figure 3E and F*). As illustrated in *Figure 3E and F*, chemogenetic inhibition of VGAT-positive ZI neurons resulted in robust bilateral hypersensitivity to tactile and pressure stimulation in sham mice as well as contralateral hypersensitivity in cuff-implanted mice. Thus, compared

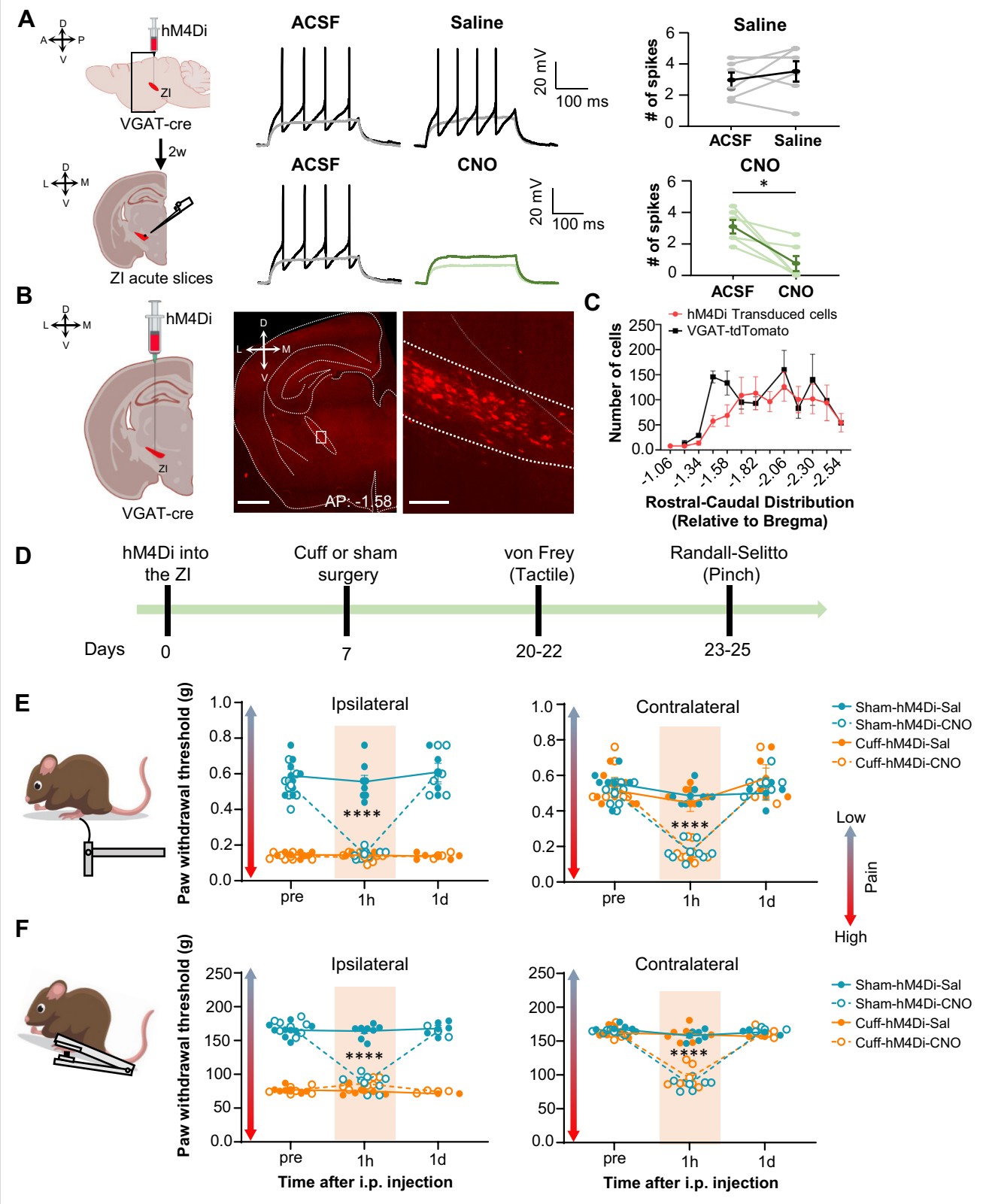

**Figure 3.** Inhibition of GABAergic ZI neurons is sufficient to induce bilateral tactile and pressure hypersensitivity in uninjured mice. (**A**) Schematic of the experimental approach. VGAT-cre mice were stereotaxically injected with hM4Di into the ZI. Current-clamp recordings were obtained from hM4Di-positive cells in acute ZI slices 2 weeks after the injection. Representative traces of whole-cell current-clamp recordings obtained from ZI neurons transduced with hM4Di before (left) and after (right) bath application of 10 μM CNO (lower panel) or vehicle (top panel). Action potentials were elicited

*Figure 3 continued*

using 500ms depolarizing current injection that evoked 2–5 action potentials before the bath application. The same amplitude of depolarizing current injection was used before and after bath application. Summary graphs depicting the mean ± SEM number of spikes before and after bath treatment are shown on the right panel (n=6 neurons per treatment; paired two-tailed t-test: t=0.98, df = 5, p=0.3722 for ACSF vs saline; Wilcoxon two-tailed matched paired signed rank test: W=–21.0, *p=0.0313 for ACSF vs CNO). Scatter points represent individual cells, with darker lines indicating the mean values +/- SEM. (**B**) Schematic diagram for unilateral stereotaxic injection of hM4Di into the ZI of VGAT-cre mice. A representative image of a coronal mouse brain slice from a VGAT-cre mouse injected with hM4Di into the ZI is shown on the middle panel. The area delineated by the white rectangle in the middle panel is shown at higher magnification in the right panel, with mCherry-positive neurons shown in red. Scale bars represent 1000 μm (left) and 100 μm (right). (**C**) Mean ± SEM number of hM4Di-transduced cells and VGAT-tdTomato labeled cells in the ZI as a function of rostral-caudal level relative to bregma (n=11 mice for hM4Di-transduced neurons and 4 mice for VGAT-tdTomato neurons). (**D**) Timeline for behavioral experiments. (**D–E**) Different modalities of pain behavior test. Responses shown as mean ± SEM paw withdrawal threshold in the ipsilateral (left panel) and contralateral (right panel) hindpaws before, 1 h and 1 day after CNO or vehicle i.p. injection in cuff or sham mice stereotaxically injected with hM4Di into the ZI. Scatter points represent individual mice. Mixed-effects model followed by Dunnett's multiple comparison test was performed for analysis of all behavioral assays. (**E**) von Frey (n=8 mice per treatment; ipsilateral hindpaw: i.p. treatment: $F_{(2,42)}$ = 31.03, ****p<0.0001; sciatic nerve treatment: $F_{(3,28)}$ = 138.8, ****p<0.0001; interaction: $F_{(6,42)}$ = 25.93, ****p<0.0001; *posthoc*: 95.00% CI of diff.=0.3129–0.4571, ****p<0.0001 for pre-injections vs 1 hr after CNO in sham-hM4Di mice; contralateral hindpaw: i.p. treatment: $F_{(2,70)}$ = 58.0, ****p<0.0001; sciatic nerve treatment: $F_{(3,70)}$ = 7.398, ***p<0.001; interaction: $F_{(6,70)}$ = 9.426, ****p<0.0001; *posthoc:* 95.00% CI of diff.=0.2276–0.4309, ****p<0.0001 for pre-injections vs 1 hr after CNO in sham-hM4Di mice and 95.00% CI of diff.=0.2787–0.4828, ****p<0.0001 for pre-injections vs 1 hr after CNO in cuff-hM4Di mice) (**F**) Randall-Selitto (ipsilateral hindpaw: n=8 for sham mice and 6 for cuff mice; i.p. treatment: $F_{(2,33)}$ = 41.16, ****p<0.0001; sciatic nerve treatment: $F_{(3,24)}$ = 265.4, ****p<0.0001; interaction: $F_{(6,33)}$ = 60.75, ****p<0.0001; *posthoc:* 95.00% CI of diff.=73.69–92.20, ****p<0.0001 for pre-injections vs 1 hr after CNO in sham-hM4Di mice; contralateral hindpaw: n=6 for sham mice and 7 for cuff mice; i.p. treatment: $F_{(2,30)}$ = 141.8, ****p<0.0001; sciatic nerve treatment: $F_{(3,22)}$ = 22.05, ****p<0.0001; interaction: $F_{(6,30)}$ = 37.21, ****p<0.0001; *Posthoc:* 95.00% CI of diff.=65.21–89.19, ****p<0.0001 for pre-injections vs 1 hr after CNO in sham-hM4Di mice and 95.00% CI of diff.=57.73–79.89, ****p<0.0001 for pre-injections vs 1 hr after CNO in cuff-hM4Di mice). See *Figure 3—figure supplement 1*.

The online version of this article includes the following source data and figure supplement(s) for figure 3:

**Source data 1.** Source data for chemogenetic inhibition of ZI-GABAergic neurons.

**Figure supplement 1.** Responses to thermal stimuli are unaltered by chemogenetic inhibition of ZI-GABAergic neurons.

**Figure supplement 1—source data 1.** Source data for thermal responses to chemogenetic inhibition of ZI-GABAergic neurons.

to pre-injection values, bilateral paw withdrawal thresholds in response to tactile or pinch stimulation were significantly (p<0.0001) reduced bilaterally following CNO injections in sham mice. Similarly, withdrawal thresholds in the hindpaw contralateral to cuff implantation were significantly (p<0.0001) reduced after CNO injection, compared to pre-injection thresholds.

Withdrawal thresholds in the hindpaw ipsilateral to cuff treatment were indistinguishable before and after chemogenetic inactivation of VGAT-positive ZI neurons, demonstrating that inhibition of ZI cells does not measurably affect cuff-induced hypersensitivity to tactile and pressure stimulation. Notably, following the chemogenetic inhibition of ZI VGAT-positive neurons, paw withdrawal thresholds in sham mice and the contralateral hindpaw of cuff-implanted mice were comparable to the withdrawal thresholds of the ipsilateral hindpaw of cuff-implanted mice. These results demonstrate that inhibition of ZI VGAT-positive neurons is sufficient to elicit hypersensitivity in the absence of injury that resembles the hypersensitivity observed following sciatic nerve cuff implantation. The effects of chemogenetic inhibition of VGAT-positive ZI neurons on tactile and pressure sensitivity were transient, as paw withdrawal thresholds in all treated animals returned to pre-injection values 1 day following CNO injections. Importantly, tactile and pressure sensitivity was unaltered in saline-injected mice, demonstrating that the CNO-induced effects were not due to handling or hM4Di expression (*Figure 3E and F*).

Previous studies have demonstrated that modulation of pain-related behaviors in the CeA, including by CeA-PKCδ neurons, is modality-dependent (*Wilson et al., 2019*). In order to evaluate whether modulation of hypersensitivity in the ZI is also modality-specific, the next set of experiments assessed the effects of inhibition of ZI VGAT-positive neurons on heat and cold sensitivity in both sham and cuff-implanted mice using the Hargreaves and acetone evaporation tests, respectively. As illustrated in *Figure 3—figure supplement 1B-D*, cuff implantation on the sciatic nerve resulted in hypersensitivity to both heat and cold stimulation in the hindpaw ipsilateral to cuff implantation compared to the contralateral hindpaw or the hindpaws of sham treated mice. Behavioral responses to cold and heat stimulation of the hindpaws, however, were unaltered by chemogenetic inhibition of VGAT-positive neurons in the ZI in all animals tested. Taken together, the results from these chemogenetic

experiments demonstrate that inhibition of VGAT-positive neurons in the ZI is sufficient to induce hypersensitivity in the absence of injury in a modality-specific manner.

## Activation of ZI-GABAergic neurons reverses cuff-induced hypersensitivity to pinch but not thermal stimulation

The next set of experiments aimed to determine whether activation of GABAergic ZI neurons is sufficient to reverse cuff-induced hypersensitivity. We again used a chemogenetic approach coupled with behavioral assays to measure tactile and heat sensitivity in mice following the implantation of a sciatic nerve cuff. Histological verification of the injection sites at the end of the experiments demonstrated that transduction of the excitatory DREADD hM3Dq and control-mCherry was restricted to the ZI (*Figure 4A* and *Figure 4—figure supplements 1 and 2A*). The numbers and rostral-caudal distribution of transduced cells within the ZI were robust and comparable between mice injected with hM3Dq and control-mCherry (*Figure 4B*). We validated CNO-mediated activation of ZI neurons in VGAT-cre mice with immunohistochemical monitoring of c-Fos, the product of an immediate early gene that is commonly used as a marker of neuronal activity (*Figure 4C*). As illustrated in *Figure 4D and i.*p. injection of CNO resulted in robust c-Fos expression in the ZI of VGAT-cre animals injected with hM3Dq, compared to the c-Fos expression observed in the ZI of saline-injected control mice that also expressed hM3Dq or CNO-injected control VGAT-cre mice stereotaxically injected with the control virus. Quantification of ZI cells co-expressing c-Fos and mCherry further confirmed that c-Fos expression in hM3Dq-transduced cells is significantly (p<0.05) higher in CNO-treated mice than in saline-treated mice or in mCherry-transduced neurons from CNO-treated mice (*Figure 4E*).

The effect of chemogenetic activation of VGAT-positive ZI cells on behavioral responses to pressure (pinching) stimulation of the hindpaws was measured before and after i.p. injection of CNO or saline in cuff and sham animals (*Figure 4F*). CNO-mediated activation of VGAT-positive ZI neurons led to significant (p<0.0001) reversal of cuff-induced hypersensitivity in the hindpaw ipsilateral to cuff implantation, while no measurable effects were seen in saline-injected or CNO-injected mCherry-control mice. The effects of activation of VGAT-positive ZI neurons were specific to nerve injury as withdrawal thresholds in sham-treated mice or the hindpaw contralateral to sciatic nerve cuff were comparable between groups. As illustrated in *Figure 4—figure supplement 2B-D*, reversal of cuff-induced hypersensitivity was also modality-specific as behavioral responses to cold and heat stimulation of the hindpaws were unaffected by chemogenetic activation of VGAT-positive neurons in the ZI. Taken together, these results demonstrate that activation of VGAT-positive neurons in the ZI reverses cuff-induced pain hypersensitivity in a modality-specific manner.

## Optogenetic inhibition of CeA-PKCδ terminals in ZI reverses cuff-induced hypersensitivity

The combined results from the anatomical, electrophysiological, and behavioral experiments described above support the hypothesis that injury-induced activation of CeA-PKCδ neurons that project to the ZI contributes to inhibition of the ZI and subsequent pain-related sensitization (*Figure 1E*). To evaluate the functional contribution of the CeA-PKCδ to ZI pathway in the modulation of injury-induced hypersensitivity, we injected a virus expressing the cre-dependent inhibitory opsin halorhodopsin (NpHR-mCherry) or a control virus expressing the mCherry fluorophore (control-mCherry) into the CeA of PKCδ-cre mice while simultaneously implanting a fiber optic probe over the ZI (*Figure 5A–B*). This approach allowed us to evaluate the effects of optically inhibiting CeA-PKCδ terminals in the ZI on nociceptive responses to pinch, tactile, heat, and cold stimulation of the hindpaws in cuff-implanted mice.

As illustrated in *Figure 5C* and *Figure 5—figure supplement 1*, these experimental manipulations resulted in robust transduction of NpHR-mCherry or control-mCherry that was restricted to the CeA, with fiber optic probes located on or right above the ventral ZI (*Figure 5D*). Whole-cell current-clamp recordings of neurons expressing NpHR-mCherry in acute CeA slices confirmed that yellow light (590 nm, 10 mW) stimulation significantly (p<0.0001) inhibited firing responses in NpHR-positive CeA cells but had no measurable effect in neighboring NpHR-negative neurons (*Figure 5E*), further validating our experimental approach.

At the behavioral level, optogenetic inhibition of CeA-PKCδ terminals in the ZI with yellow light (561 nm, 10 mW) significantly (p<0.0001) reduced cuff-induced hypersensitivity to pinch stimulation

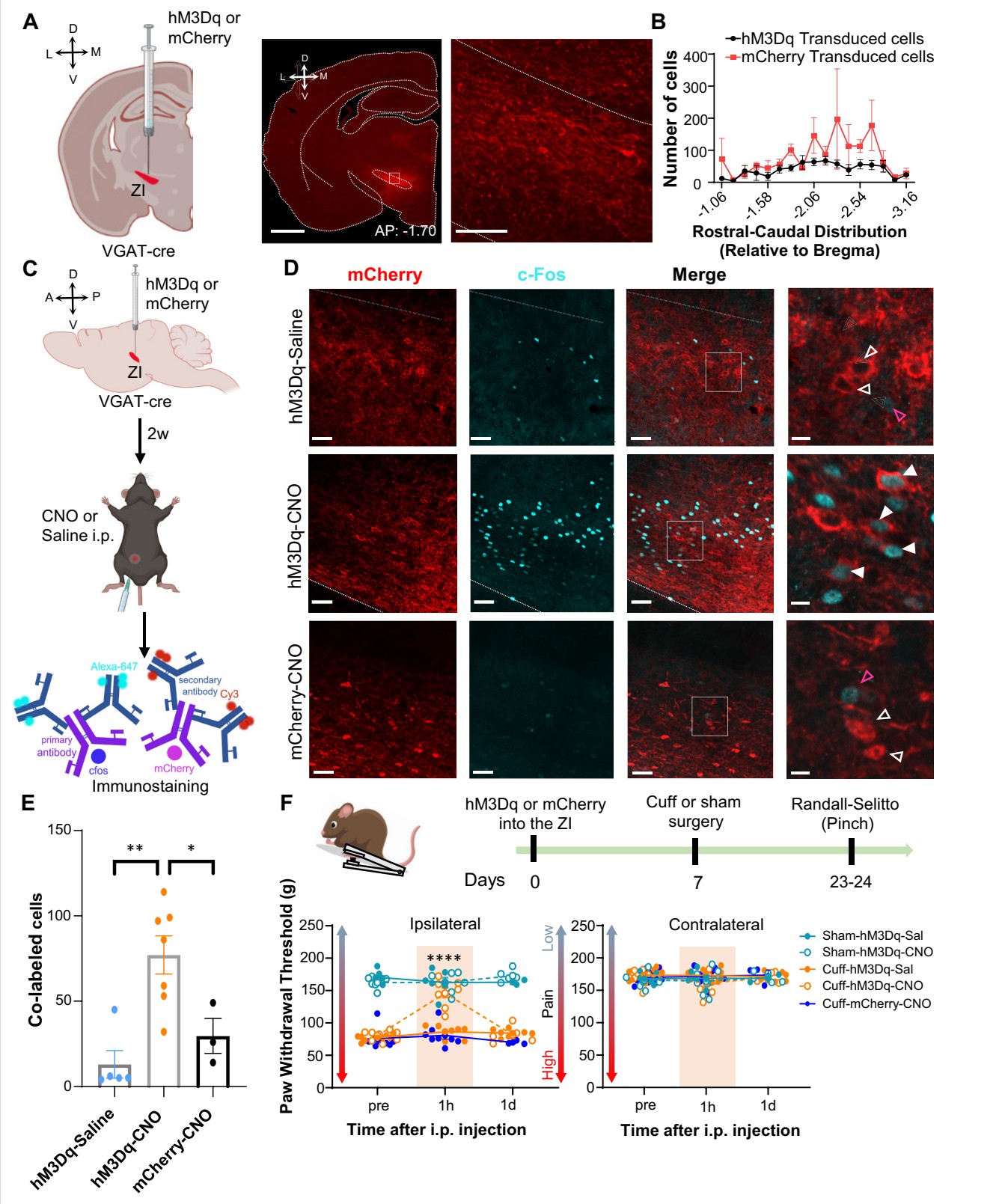

**Figure 4.** Activation of GABAergic ZI neurons reverses cuff-induced hypersensitivity to pinch stimulation. (**A**) VGAT-cre mice were injected with hM3Dq or mCherry into the ZI. Low-magnification representative image of a coronal brain slice shows the site of virus injection in red. The area delineated by the white rectangle is shown at higher magnification in the right image. Scale bars are 1000 µm for low magnification and 100 µm for high-magnification images (**B**) Quantification of ZI cells transduced with hM3Dq and mCherry is shown as mean ± SEM (n=17 mice for hM3Dq transduced group and 7 mice

*Figure 4 continued on next page*

*Figure 4 continued*

for mCherry group). (**C**) c-Fos experimental timeline. (**D**) Representative images of coronal brain slices containing the ZI of VGAT-cre mice injected with hM3Dq (top and middle panels) or mCherry (bottom) into the ZI and i.p. treated with CNO (middle and bottom panels) or saline (top panel). mCherry expression is shown in red and immunostaining for c-Fos in cyan. The merged images are shown in the rightmost panels. White boxes delineate the areas magnified on the right panel. Magenta open arrowheads point to cells that are positive for c-Fos only; white open arrowheads point to cells that are positive for mCherry only; solid arrowheads point to cells that are positive for both mCherry and c-Fos. Scale bars are 50 µm (low magnification) and 10 µm (high magnification). (**E**) Mean ± SEM numbers of c-Fos and mCherry transduced co-labeled cells per condition. (n=7 for hM3Dq-CNO, 5 for hM3Dq-saline and 3 for mCherry-CNO mice; One-way ANOVA followed by Tukey's multiple comparisons test: $F_{(2,12)}$ = 10.90; **p<0.01 for hM3Dq-CNO vs hM3Dq-Sal and *p<0.05 for hM3Dq-CNO vs mCherry-CNO). (**F**) Randall-Selitto responses are shown as mean ± SEM paw withdrawal threshold in the ipsilateral (left panel) and contralateral (right panel) hindpaw before, 1 hr and 1 day after CNO or vehicle i.p. injections in cuff or sham mice stereotaxically injected with hM3Dq or mCherry into the ZI (n=6 for sham- hM3Dq mice, n=8 for cuff-hM3Dq and mCherry-cuff mice; Mixed-effects model followed by Dunnett's multiple comparison test; Ipsilateral hindpaw: i.p. treatment: $F_{(2,48)}$ = 16.99, ****p<0.0001; sciatic nerve treatment: $F_{(4,31)}$ = 112.9, ****p<0.0001; interaction: $F_{(8,48)}$ = 17.52, ****p<0.0001; *posthoc:* 95.00% CI of diff.=–77.97 to –53.12, ****p<0.0001 for pre-injections vs 1 hr after CNO in cuff-hM3Dq mice; contralateral hindpaw: i.p. treatment: $F_{(2,48)}$ = 0.9387, p=0.3982; sciatic nerve treatment: $F_{(4,31)}$ = 0.4981, p=0.7372; interaction: $F_{(8,48)}$ = 1.070, p=0.3996). Scatter points represent individual mice. See *Figure 4—figure supplements 1 and 2*.

The online version of this article includes the following source data and figure supplement(s) for figure 4:

**Source data 1.** Source data for chemogenetic activation of ZI-GABAergic neurons.

**Figure supplement 1.** Rostral-caudal distribution of hM3Dq injection sites in mice used for behavioral experiments.

**Figure supplement 2.** Cuff-induced thermal hypersensitivity is unaltered by chemogenetic activation of ZI-GABAergic neurons.

**Figure supplement 2—source data 1.** Source data for thermal responses to chemogenetic activation of ZI-GABAergic neurons in cuff implanted mice.

of the treated paw without affecting the untreated paw (*Figure 5F*). Yellow light-mediated behavioral effects were dependent on laser intensity, with increasing laser intensities resulting in larger reductions of cuff-induced hypersensitivity at intensities between 0.04 and 0.06 mW, and with a plateau observed at intensities equal to or higher than 0.07 mW (*Figure 5G*).

Based on these results, we used 0.07 mW and 10 mW yellow light intensities for all subsequent nociceptive behavioral assays. As illustrated in *Figure 6A–D*, optogenetic inhibition of CeA-PKCδ terminals in ZI resulted in significant (p<0.0001) reductions of cuff-induced hypersensitivity to tactile (*Figure 6B*), cold (*Figure 6C*), and heat (*Figure 6D*) stimulation of the treated hindpaw at both 0.07 mW and 10 mW laser intensities. The laser-induced behavioral effects were specific to NpHR expression as we did not observe measurable differences in behavioral responses of mice injected with the control-mCherry virus (*Figure 5F–G* and *Figure 6*).

Altogether, these results indicate that injury-induced activation of the CeA-PKCδ to ZI inhibitory pathway is necessary for cuff-induced pressure, tactile, and thermal hypersensitivity.

## Optogenetic activation of CeA-PKCδ terminals in ZI promotes pain-related hypersensitivity in the uninjured paw

Previous studies have shown that activation of CeA-PKCδ neurons induces bilateral tactile hypersensitivity in the absence of injury *Wilson et al., 2019*. In the present study, we further showed that inhibition of GABAergic ZI neurons also induces bilateral tactile hypersensitivity (*Figure 3*), suggesting that CeA-PKCδ-mediated inhibition of the ZI drives tactile hypersensitivity. In the next experiments, we tested this hypothesis directly by injecting a virus expressing the cre-dependent ChR2-mCherry or a control virus expressing the mCherry fluorophore (control-mCherry) into the CeA of PKCδ-cre mice while simultaneously implanting a fiber optic probe over the ZI (*Figure 7A–B*).

As illustrated in *Figure 7C* and *Figure 5—figure supplement 1*, this approach resulted in robust transduction of ChR2-mCherry or control-mCherry that was restricted to the CeA, with fiber optic probes located on or right above the ventral ZI (*Figure 7D*). Whole-cell current-clamp recordings of neurons expressing ChR2-mCherry in acute CeA slices further confirmed that blue light (470 nm, 10 mW) stimulation significantly (P<0.0001) increased firing responses in ChR2-positive CeA cells but had no measurable effect in control ChR2-negative neighboring neurons (*Figure 7E–G*).

Optogenetic activation of CeA-PKCδ terminals in ZI with blue light (473 nm, 10 mW) significantly (p<0.0001) increased sensitivity to pinch stimulation of the untreated hindpaw without affecting responses to stimulation of the treated hindpaw or responses in PKCδ-cre mice injected with the control-mCherry virus (*Figure 7H*). Light intensity-response experiments showed that behavioral responses to blue light stimulation are intensity-dependent. Thus, light-induced increases in the

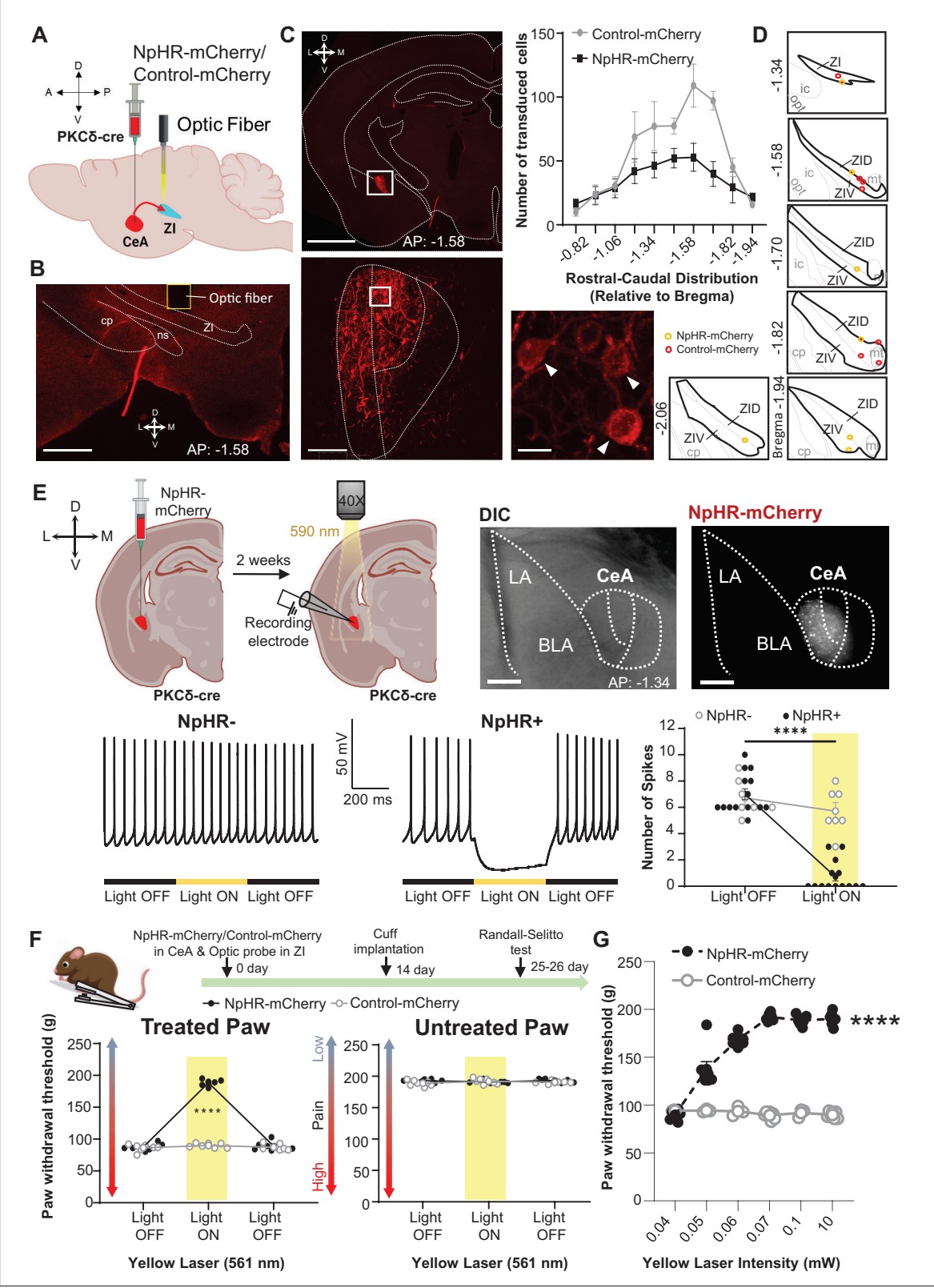

**Figure 5.** Optogenetic inhibition of CeA-PKCδ terminals in ZI reverses cuff-induced hypersensitivity. (**A**) Schematic of experimental approach. PKCδ-cre mice were stereotaxically injected with NpHR-mCherry or control-mCherry virus into the CeA and simultaneously implanted with an optic fiber above the ZI. (**B**) Representative image of coronal brain slice illustrating the placement of the optic fiber above the ZI (scale bar 500 μm). (**C**) Top left panel shows representative image of coronal brain slice depicting anatomical location of NpHR-mCherry transduced cells in the CeA (scale bar 1 mm).

*Figure 5 continued on next page*

*Figure 5 continued*

A higher magnification of the CeA, delineated by the white rectangle, is shown in the bottom left panel. NpHR-positive CeA neurons are highlighted by white arrowheads in the inset shown in right bottom. Scale bars represent 100 μm for left bottom image and 10 μm for right bottom image. Mean ± SEM number of NpHR-transduced and control-mCherry-transduced CeA cells as a function of rostral-caudal level relative to bregma are shown in the top right panel (n=6 mice for NpHR-transduced cells and n=6 mice for control-mCherry cells). (**D**) Drawing maps illustrating location of optic fiber tips in animals used for opto-inhibition study. Symbol (**O**) indicates where the cannula tips were placed in the ZI area. ZID = dorsal zona incerta, ZIV = ventral zona incerta, ic = internal capsule, cp = cerebral peduncle, opt = optic tract, mt = mammillothalamic tract. (**E**) Top left panel – Schematic diagram of experimental approach. NpHR-mCherry was injected in the CeA of PKCδ-cre mice. Whole-cell current-clamp recordings of NpHR-negative and NpHR-positive neurons in acute CeA slices were performed 2 weeks following the stereotaxic injection. Top right panel – representative differential contrast and fluorescent images of a coronal brain slice illustrating the anatomical localization of NpHR-transduced cells in the CeA. Scale bars represent 500 μm. LA = lateral amygdala, BLA = basolateral amygdala, CeA = central amygdala. Bottom left panel – representative voltage traces from NpHR-negative and NpHR-positive neurons before, during and after yellow light ($\lambda$ =590 nm) stimulation. Black lines represent light off and yellow lines represent light on. Bottom right panel –mean ± SEM number of action potentials before and during light stimulation of NpHR-negative and NpHR-positive neurons (n=7 NpHR-negative and 14 NpHR-positive neurons; two-way ANOVA followed by Šídák's multiple comparisons test: $F_{(1,19)}$ = 72.57, ****p<0.0001 for light off vs light on in NpHR-positive neurons). Scatter points represent individual cells. (**F**) Top panel shows experimental timeline. Randall-Selitto responses are shown as mean ± SEM paw withdrawal threshold in the cuff-treated (left panel) and untreated (right panel) hindpaw before, during and after yellow ($\lambda$ =561 nm, 10 mW) light-induced inhibition of CeA-PKCδ terminals in the ZI in mice expressing NpHR-mCherry or control-mCherry in the CeA (n=6 for NpHR-mCherry mice and 7 for control-mCherry mice; two-way repeated measures ANOVA followed by Dunnett's multiple comparison test: treated hindpaw: light treatment: $F_{(2,22)}$=339.3, ****p<0.0001; brain treatment: $F_{(1,11)}$ = 473.0; ****p<0.0001; interaction: $F_{(2,22)}$=300.0; ****p<0.0001; *posthoc:* 95.00% CI of diff.=–109.1 to –93.22, ****p<0.0001 for light off vs light on in NpHR-mCherry mice; untreated hindpaw: light treatment: $F_{(2,22)}$ = 0.3367; p=0.7177; brain treatment: $F_{(1,11)}$ = 2.278; p=0.1594; interaction: $F_{(2,22)}$ = 2.380; p=0.1160). Yellow bar indicates the behavioral response during opto-inhibition of CeA-PKCδ terminals in the ZI (**G**) Yellow laser intensity response curve during Randall-Selitto in the cuff-implanted hindpaw. Mean ± SEM paw withdrawal response as a function of increasing laser intensities during yellow ($\lambda$ =561 nm) light-induced inhibition of CeA-PKCδ terminals in the ZI of mice expressing NpHR-mCherry or control-mCherry in the CeA (n=3–7 for control-mCherry and n=7 for NpHR-mCherry mice; two-way repeated measures ANOVA: $F_{(2,22)}$=339.3, ****p<0.0001 for laser intensity effect on treated hindpaw withdrawal responses in mice expressing NpHR-mCherry). Scatter points represent individual mice. See *Figure 5—figure supplement 1*.

The online version of this article includes the following source data and figure supplement(s) for figure 5:

**Source data 1.** Source data for optogenetic inhibition of CeA-PKCδ terminals in ZI.

**Figure supplement 1.** Rostral-caudal distribution of injection sites in PKCδ-cre mice used for optogenetics behavioral experiments.

sensitivity to pinch of the untreated paw was observed at intensities equal to or higher than 0.6 mW, with a plateau observed starting at 0.8 mW (*Figure 7I*). Based on these intensity-response results, we used 0.8 mW and 10 mW for subsequent nociceptive behavior experiments.

Consistent with pinch stimulation results, optogenetic activation of CeA-PKCδ terminals in ZI with blue light at 0.8 mW or 10 mW significantly (*P*<0.01) increased sensitivity to tactile (*Figure 8B*), cold (*Figure 8C*), and heat (*Figure 8D*) stimulation of the untreated hindpaw. No measurable effects were seen in the treated hindpaw or in control virus-injected mice after blue light stimulation. These results show that activation of the CeA-PKCδ to ZI inhibitory pathway induces hypersensitivity to tactile, pressure, heat, and cold stimuli in the absence of injury.

## Discussion

Previous studies have shown that spontaneous firing rates and somatosensory-evoked neuronal responses decrease in the ZI in a rodent model of central pain syndrome (*Masri et al., 2009*). In the CeA, previous studies have demonstrated that CeA-PKCδ neurons are activated in a rodent model of neuropathic pain and drive behavioral hypersensitivity (*Wilson et al., 2019*). The results presented here demonstrate a strong inhibitory input from the CeA to the ZI (*Figure 2C–F*). Our anterograde and retrograde experiments further showed that a subset of ZI-projecting CeA neurons are CeA-PKCδ (*Figures 1–2*). Moreover, our experiments showed that inhibition of GABAergic ZI neurons induced robust bilateral behavioral hypersensitivity whereas activation of these ZI cells reversed injury-induced hypersensitivity (*Figures 3–4*). At the circuit level, we showed that inhibitory inputs from CeA-PKCδ neurons to ZI are necessary for cuff-induced hypersensitivity whereas activation of this pathway induces hypersensitivity to tactile, pressure, heat, and cold stimuli in the absence of injury (*Figures 5–8*). Together, our study identified a new inhibitory neural circuit from CeA-PKCδ neurons to ZI where injury-induced activation of CeA-PKCδ neurons inhibits ZI-GABAergic cells, subsequently leading to behavioral hypersensitivity (*Figure 9*).

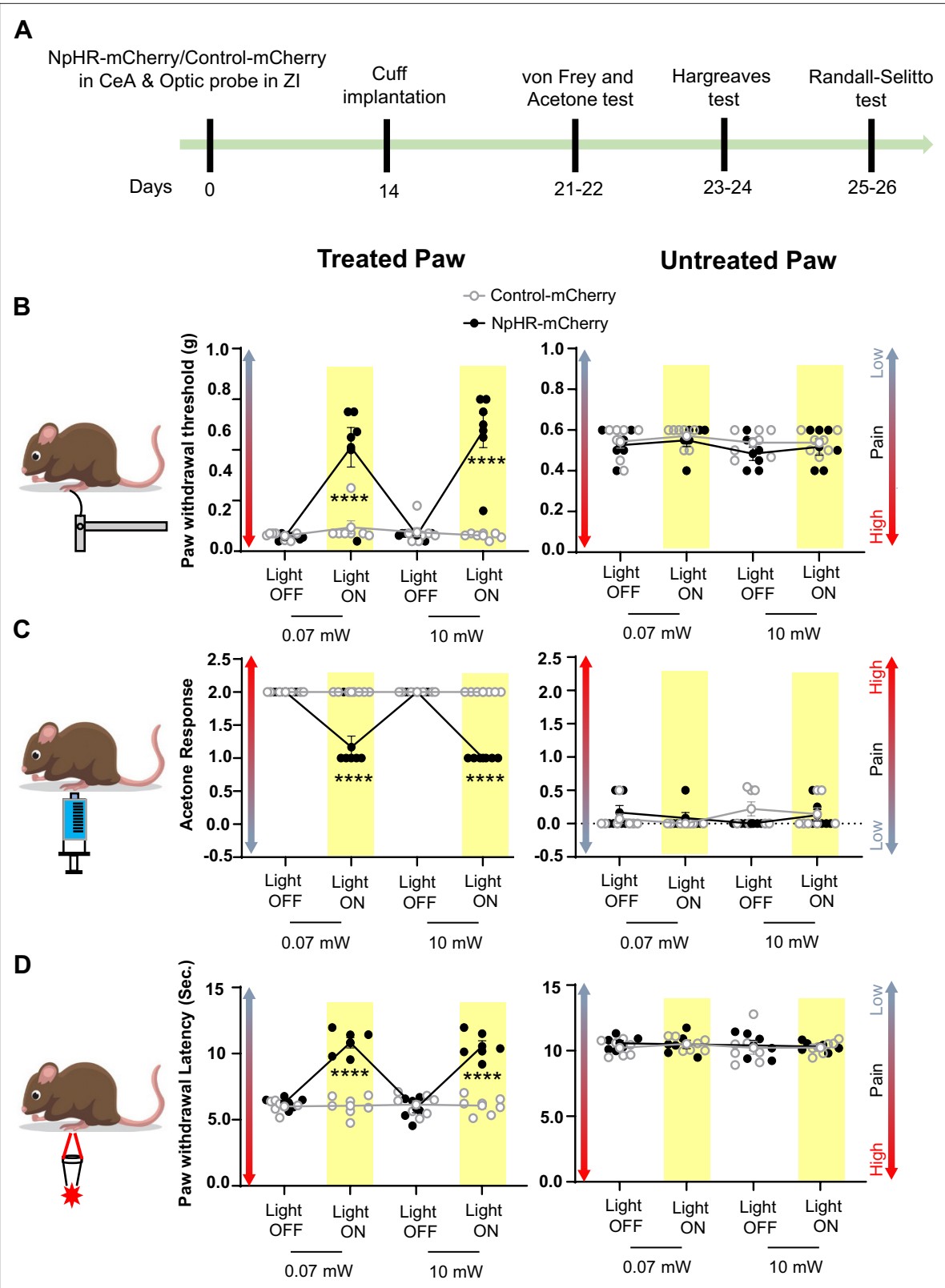

**Figure 6.** Inhibition of CeA-PKCδ terminals in ZI reduces cuff-induced tactile and thermal hypersensitivity. (**A**) Experimental timeline of viral infection in CeA, optic probe implantation in ZI, cuff implantation in sciatic nerve and battery of nociceptive behavior tests. (**B–D**) Different modalities of pain related behavioral tests in cuff-treated (left panel) and untreated (right panel) hindpaws before and during yellow ($\lambda$ =561 nm, 0.07 mW and 10 mW) light-induced inhibition of CeA-PKCδ terminals in the ZI of mice expressing NpHR-mCherry or control-mCherry in the CeA. Two-way repeated measures

*Figure 6 continued on next page*

*Figure 6 continued*

ANOVA followed by Dunnett's multiple comparison test was performed for all behavioral assays. (**B**) Tactile hypersensitivity using von Frey test (n=6 for NpHR-mCherry mice and 7 for control-mCherry mice; treated hindpaw: light treatment: $F_{(3,33)}$ = 28.79; ****p<0.0001; brain treatment: $F_{(1,11)}$ = 26.42; ****p<0.0001; interaction: $F_{(3,33)}$ = 26.71; ****p<0.0001; *posthoc*: 95.00% CI of diff.=–0.4626 to –0.2456, ****p<0.0001 for 0.07 mW and 95.00% CI of diff.=–0.5289 to –0.3119, ****p<0.0001 for 10 mW light intensity for light off vs light on in NpHR-mCherry mice; untreated hindpaw: light treatment: $F_{(3,33)}$ = 0.8907; p=0.4561; brain treatment: $F_{(1,11)}$ = 3.723; p=0.0798; interaction: $F_{(3,33)}$ = 0.1554; p=0.9255) (**C**) cold hypersensitivity using acetone test (n=6 for NpHR-mCherry mice and 7 for control-mCherry mice; treated paw: light treatment: $F_{(3,33)}$ = 48.57; ****p<0.0001; brain treatment: $F_{(1,11)}$ = 143.3; ****p<0.0001; interaction: $F_{(3,33)}$ = 48.57; ****p<0.0001; *posthoc*: 95.00% CI of diff.=0.6377–1.029, ****p<0.0001 for 0.07 mW and 95.00% CI of diff.=0.8044–1.196, ****p<0.0001 for 10 mW light intensity for light off vs light on in NpHR-mCherry mice; untreated hindpaw: light treatment: $F_{(3,33)}$ = 0.7780; p=0.5147; brain treatment: $F_{(1,11)}$=0.03751; p=0.8500; interaction: $F_{(3,33)}$ = 2.495; p=0.0770) and (**D**) heat hypersensitivity using Hargreaves test (n=6 for NpHR-mCherry mice and 7 for control-mCherry mice; treated hindpaw: light treatment: $F_{(3,33)}$ = 40.88; ****p<0.0001; brain treatment: $F_{(1,11)}$ = 122.3, ****p<0.0001; interaction: $F_{(3,33)}$ = 41.89; ****p<0.0001; posthoc: 95.00% CI of diff.=–5.619 to –3.49, ****p<0.0001 for 0.07 mW and 95.00% CI of diff.=–5.353 to –3.227, ****p<0.0001 for 10 mW light intensity for light off vs light on in NpHR-mCherry mice; untretaed hindpaw: light treatment: $F_{(3,33)}$ = 0.2300; p=0.8748; brain treatment: $F_{(1,11)}$ = 0.4414; p=0.5201; interaction: $F_{(3,33)}$ = 0.1591, p=0.9231). Yellow bar indicates the behavioral response during opto-inhibition of CeA-PKCδ terminals in the ZI. Results are expressed as mean ± SEM. Scatter points represent individual mice.

The online version of this article includes the following source data for figure 6:

**Source data 1.** Source data for behavioral responses to tactile, cold and heat stimulation after optogenetic inhibition of CeA-PKCδ terminals in ZI.

## Central Amygdala inputs to the Zona Incerta

Using AAVs expressing mCherry or ChrimsonR-tdTomato, we identified 18 brain regions that receive efferent projections from CeA-PKCδ neurons (*Figure 1* and *Table 1*). The results were comparable between injected brains independently of AAV used and all 18 brain regions identified have been previously defined as output regions of the CeA using traditional anterograde tracers (*Aggleton, 2000*; *Barbier et al., 2017*; *Reardon and Mitrofanis, 2000*; *Shinonaga et al., 1992*; *Zhou et al., 2018*), validating our experimental approach to study the efferent projections of CeA-PKCδ neurons. Our qualitative and quantitative anatomical analyses further showed moderate labeling of CeA-PKCδ terminals that are dependent on the subregions and rostral-caudal level of the ZI (*Figure 1—figure supplement 2*).

Follow up retrograde experiments showed uptake in both PKCδ-positive and PKCδ-negative cells in the CeA when the retrograde tracer was injected in the ZI (*Figure 2B* and *Figure 2—figure supplement 1*). These results are consistent with previous studies that have shown that the majority of ZI-projecting CeA neurons are somatostatin-expressing (*Zhou et al., 2018*), which have virtually no overlap with CeA-PKCδ neurons (*Adke et al., 2021*). Our anatomical experiments clearly show, however, a moderate density of CeA-PKCδ axonal terminals that is anatomically restricted to the middle and ventral sectors of the ZI (*Figure 1—figure supplement 2*) and originates in the capsular and lateral subdivisions of the middle and posterior CeA (*Figure 2—figure supplement 1*). Notably, uptake of the retrograde tracer in PKCδ-negative neurons was readily evident in the medial subdivision and rostral sectors of the CeA, which correspond to CeA regions previously shown to have high densities of somatostatin-positive neurons (*Adke et al., 2021*). Based on these combined findings, we hypothesize that cell-type and sub-nuclei-specific CeA inputs are anatomically segregated in the ZI and might differentially contribute to modulation of behavioral output.

## Functional contribution of CeA-PKCδ to ZI pathway in nociceptive behaviors

Previous studies have reported that the ZI is inhibited in a rat model of central pain syndrome (*Masri et al., 2009*). The source of inhibitory input to the ZI, however, remained unknown. In the present study, we showed that the ZI receives moderate axonal terminals from CeA-PKCδ neurons (*Figure 1—figure supplement 2*) and that inhibitory inputs from CeA-PKCδ neurons to the ZI contribute to persistent pain-related behaviors. It is thus reasonable to postulate that CeA-PKCδ neurons serve as an inhibitory input to the ZI that contributes to the regulation of nociceptive behaviors and provides a mechanistic explanation for the previously reported reductions in ZI activity during pain-related states (*Masri et al., 2009*).

At the behavioral level, we showed that optogenetic inhibition of CeA-PKCδ terminals in the ZI reduced hypersensitivity to pinch, tactile, cold and heat stimulation (*Figures 5–6*). These results are consistent with previous studies showing that chemogenetic inhibition of CeA-PKCδ neurons also

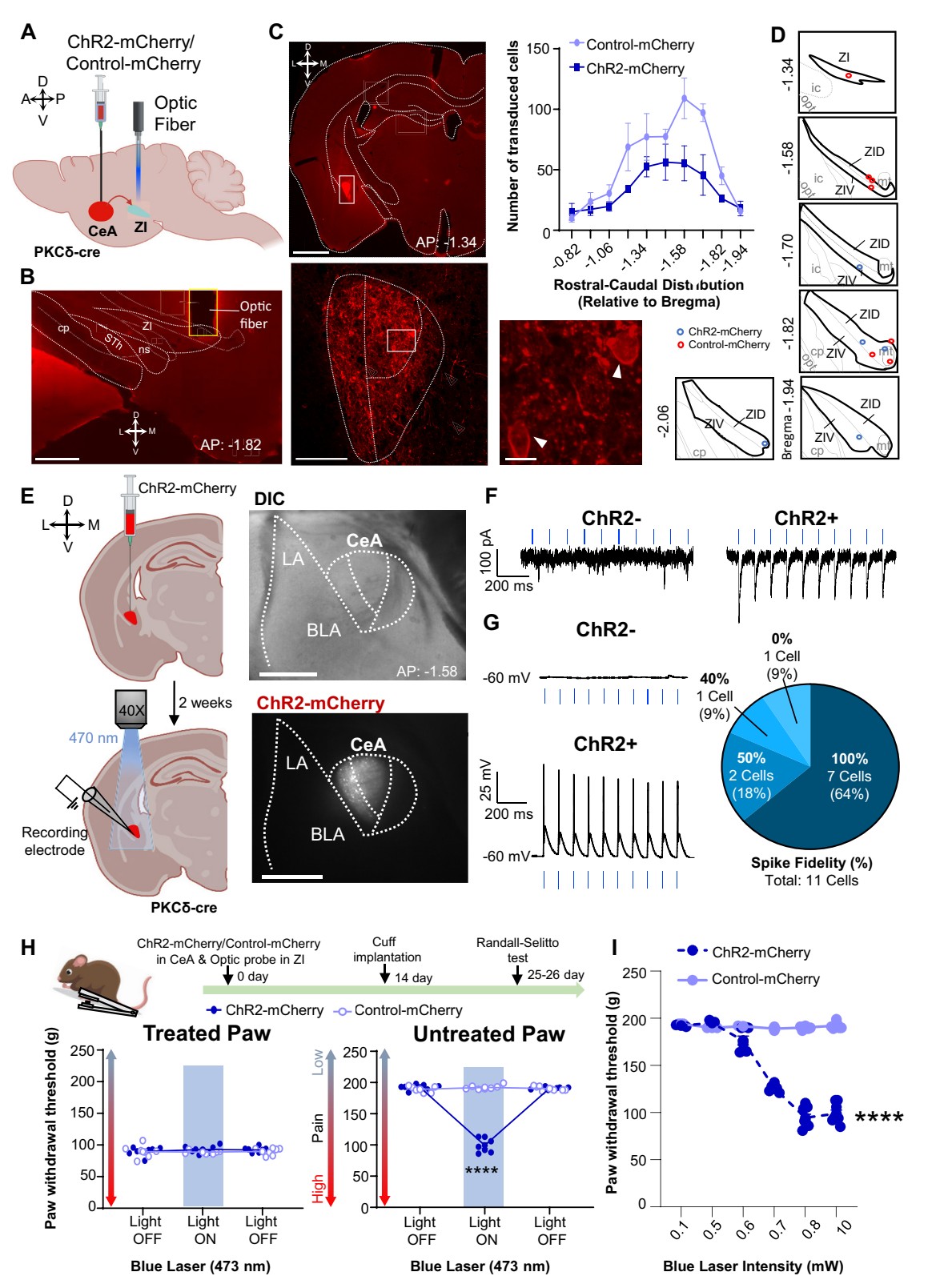

**Figure 7.** Optogenetic activation of CeA-PKCδ terminals in ZI induces pain related hypersensitivity in uninjured paw. (**A**) Schematic of experimental approach. PKCδ-cre mice were stereotaxically injected with ChR2-mCherry or control-mCherry virus into the CeA and simultaneously implanted with an optic fiber above the ZI. (**B**) Representative image of coronal brain slice showing the placement of the optic fiber above the ZI (scale bar 500 μm). (**C**) Top left panel shows representative image of coronal brain slice illustrating anatomical localization of ChR2-mCherry-transduced cells in the CeA (scale bar

*Figure 7 continued on next page*

*Figure 7 continued*

1 mm). The area delineated by the white box is shown at higher magnification in the bottom left panel. ChR2-positive neurons are highlighted by white arrowheads in the insets shown in the right bottom panel. Scale bars represent 100 µm for left bottom image and 10 µm for right bottom image. Mean ± SEM number of ChR2-transduced and control-mCherry transduced CeA cells as a function of rostral-caudal level relative to bregma are shown in the top right panel (n=4 mice for ChR2-transduced neurons and n=6 mice for control-mCherry neurons). (**D**) Drawing maps illustrating location of optic fiber tips in animals used for opto-activation study. Symbol (**O**) indicates where the cannula tips were placed in the ZI area. ZID = dorsal zona incerta, ZIV = ventral zona incerta, ic = internal capsule, cp = cerebral peduncle, opt = optic tract, mt = mammillothalamic tract. (**E**) Left panel - schematic diagram of experimental approach. ChR2-mCherry was injected in the CeA of PKCδ-cre mice. Patch-clamp recordings of ChR2-negative and ChR2-positive neurons in acute brain slices were collected 2 weeks following the stereotaxic injection. Right panel – representative differential contrast and fluorescent images illustrating the anatomical localization of ChR2-transduced cells in the CeA. Scale bars represent 500 µm. LA = lateral amygdala, BLA = basolateral amygdala, CeA = central amygdala. (**F–G**) Representative current (**F**) and voltage (**G**) traces of ChR2-negative and ChR2-positive neurons in response to blue light ($\lambda$ =470 nm, 10 Hz, 5ms) stimulation, depicted by the blue bars. Right panel in (**G**) proportion of ChR2-positive neurons with different spike fidelity (n=11 ChR2-positive cells, n=3 ChR2-negative cells). (**H**) Top panel shows experimental timeline. Randall-Selitto responses are shown as mean ± SEM paw withdrawal threshold in cuff-treated (left panel) and untreated (right panel) hindpaws before, during and after blue ($\lambda$ =473 nm, 10 mW) light-induced activation of CeA-PKCδ terminals in the ZI of mice expressing ChR2-mCherry or control-mCherry in the CeA (n=7 for both ChR2-mCherry and control-mCherry mice; two-way repeated measures ANOVA followed by Dunnett's multiple comparison test: treated hindpaw- light treatment: $F_{(2, 24)}$=0.2333; p=0.7937; brain treatment: $F_{(2, 12)}$ = 0.8852; p=0.3653; interaction: $F_{(2, 24)}$=0.4450; p=0.6460; untreated hindpaw: light treatment: $F_{(2,24)}$ = 314.7; ****p<0.0001; brain treatment: $F_{(1,12)}$ = 220.6; ****p<0.0001; interaction: $F_{(2,24)}$ = 352.0; ****p<0.0001; *posthoc:* 95.00% CI of diff.=86.28–100.0, ****p<0.0001 for light off vs light on in ChR2-mCherry mice). Blue bar indicates the behavioral response upon opto-activation of CeA-PKCδ terminals in ZI. Scatter points represent individual mice. (**I**) Blue ($\lambda$ =473 nm) laser intensity response curve during Randall-Selitto in the untreated hindpaw. Mean ± SEM paw withdrawal response as a function of increasing laser intensities during blue ($\lambda$ =473 nm) light induced activation of CeA-PKCδ terminals in the ZI of mice expressing ChR2-mCherry or control-mCherry in the CeA (n=3–8 for ChR2-mCherry mice and n=2–7 for control-mCherry mice; two-way repeated measures ANOVA: $F_{(5, 31)}$=77.96, ****p<0.0001 for laser intensity effect on untreated paw withdrawal responses in mice expressing ChR2-mCherry). Scatter points represent individual mice.

The online version of this article includes the following source data for figure 7:

**Source data 1.** Source data for optogenetic activation of CeA-PKCδ terminals in ZI.

decreases hypersensitivity to all modalities (*Wilson et al., 2019*). Notably, however, our experiments also showed that thermal modalities are unaffected by chemogenetic manipulations of ZI-VGAT neurons (*Figures 3–4*). These combined results suggest heterogenous functions of ZI neurons in the modulation of pain-related behaviors, with CeA-PKCδ neurons targeting a specific subpopulation of ZI neurons that contributes to modulation of all pain modalities. Consistent with this idea, previous studies have shown that ZI is a heterogenous structure that contains multiple cell types (*Mitrofanis, 2005*).

Heterogeneity of function in the CeA has also been previously reported (*Kim et al., 2017*; *Kong and Zweifel, 2021*; *Moscarello and Penzo, 2022*; *Venniro et al., 2020*). In the context of pain, CeA-PKCδ and somatostatin-positive CeA neurons have been shown to have opposing functions in the modulation of pain-related responses (*Wilson et al., 2019*). Our results indicate that CeA-PKCδ neurons provide inhibitory input to the ventral sector of ZI, and that activation of this inhibitory pathway drives behavioral hypersensitivity. The function of this pathways and cells in the modulation of spontaneous (non-evoked) pain-related behaviors and the affective component pain, however, remains unknown.

## Modulation of pain-related behaviors in the ZI

In the present study, we show that chemogenetic manipulation of the activity of VGAT-positive ZI neurons bidirectionally modulates pain-related behaviors in a modality-specific manner (*Figures 3–4*). Thus, we observe bilateral hypersensitivity to tactile (but not thermal) stimulation following inhibition of VGAT-positive ZI neurons in the absence of nerve or tissue injury (*Figure 3* and *Figure 3—figure supplement 1B-D*). In contrast, chemogenetic activation of VGAT-positive ZI neurons reversed nerve injury-induced hypersensitivity to pinch (but not thermal) stimulation (*Figure 4* and *Figure 4—figure supplement 2B-D*). Our results are consistent with previous studies that have shown that optogenetic activation or inhibition of VGAT-positive ZI neurons influences sensitivity to tactile stimulation (*Hu et al., 2019*). It is important to note that baseline responses in sham animals or the hindpaw contralateral to sciatic nerve treatment were unaltered by chemogenetic activation of ZI-VGAT neurons in the present study. This contrasts with previous reports, where optogenetic activation of ZI-VGAT neurons decreased baseline responses to tactile stimulation (*Hu et al., 2019*). Previous studies have

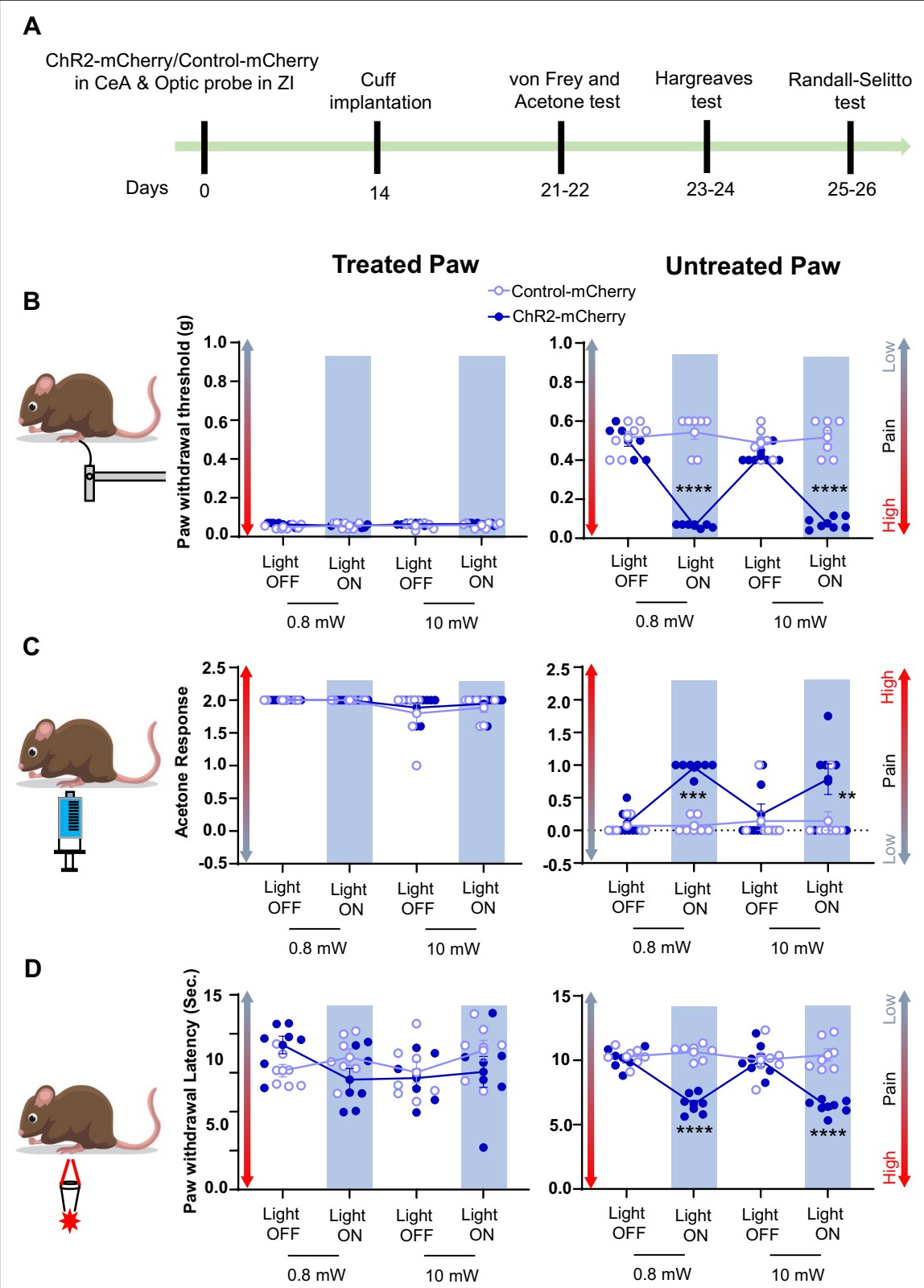

**Figure 8.** Activation of CeA-PKCδ terminals in ZI induces pain related tactile and thermal hypersensitivity in uninjured paw. (**A**) Experimental timeline of viral infection in CeA, optic probe implantation in ZI, cuff implantation in sciatic nerve and battery of nociceptive behavior tests. (**B–D**) Different modalities of pain related behavioral tests in cuff-treated (left panel) and untreated (right panel) hindpaws before and during blue light ( λ =473 nm, 0.8 mW and 10 mW) light-induced activation of CeA-PKCδ terminals in the ZI of mice expressing ChR2-mCherry or control-mCherry in the CeA. Two-way

*Figure 8 continued on next page*

*Figure 8 continued*

repeated measures ANOVA followed by Dunnett's multiple comparison test was performed for all behavioral assays. n=7 mice per treatment. (**B**) Tactile hypersensitivity using von Frey test (treated hindpaw: light treatment: $F_{(3,36)}$ = 1.287; p=0.2937; brain treatment: $F_{(1,12)}$ = 2.390; p=0.1481; interaction: $F_{(3,36)}$ = 1.511; p=0.2282; untreated hindpaw: light treatment: $F_{(3,36)}$ = 92.80; ****p<0.0001; brain treatment: $F_{(1,12)}$ = 60.38; ****p<0.0001; interaction: $F_{(3,36)}$ = 120.8; ****p<0.0001; *posthoc:* 95.00% CI of diff.=0.3811–0.4896, ****p<0.0001 for 0.8 mW and 95.00% CI of diff.=0.3693–0.4778, ****p<0.0001 for 10 mW light intensity for light off vs light on in ChR2-mCherry mice) (**C**) cold hypersensitivity using acetone test (treated hindpaw: light treatment: $F_{(3,36)}$ = 3.418; p=0.0274; brain treatment: $F_{(1,12)}$ = 0.3386; p=0.5714; interaction: $F_{(3,36)}$ = 0.2698; p=0.8467; untreated hindpaw: light treatment: $F_{(3,36)}$ = 4.760; p=0.0068; brain treatment: $F_{(1,12)}$ = 29.45; ***p=0.0002; interaction: $F_{(3,36)}$ = 4.817; p=0.0064; *posthoc:* 95.00% CI of diff.=–1.324 to –0.3905, ***p=0.0002.for 0.8 mW and 95.00% CI of diff.=–1.145 to –0.2119, **p=0.0030 for 10 mW intensity of light off vs light on in ChR2-mCherry mice) and (**D**) heat hypersensitivity using Hargreaves test (treated hindpaw: light treatment: $F_{(3,36)}$ = 0.9690; p=0.4180: brain treatment: $F_{(1,12)}$ = 1.420; p=0.2564; interaction: $F_{(3,36)}$ = 2.020; p=0.1284; untreated hindpaw: light treatment: $F_{(3,36)}$ = 12.90; ****p<0.0001; brain treatment: $F_{(1,12)}$ = 69.76; ****p<0.0001; interaction: $F_{(3,36)}$ = 19.54; ****p<0.0001; *posthoc:* 95.00% CI of diff.=2.249–4.771, ****p<0.0001 for 0.8 mW and 95.00% CI of diff.=2.451–4.972, ****p<0.0001 for 10 mW intensity of light off vs light on in ChR2-mCherry mice). Blue bar indicates the behavioral response during opto-activation of CeA-PKCδ terminals in the ZI. Results are expressed as mean ± SEM. Scatter points represent individual mice.

The online version of this article includes the following source data for figure 8:

**Source data 1.** Source data for behavioral responses to tactile, cold and heat stimulation after optogenetic inhibition of CeA-PKCδ terminals in ZI.

demonstrated that modulation of behavioral output by optogenetic stimulation is dependent on the frequency and pattern of the stimulation used for the experiments (*Padilla-Coreano et al., 2019*). A recent study further demonstrated that deep brain stimulation of zona incerta in human subjects reduced experimental heat pain without affecting mechanical pain thresholds (*Lu et al., 2021*). It is therefore possible that the differences in the results from our chemogenetic study and the previous rodent optogenetic and human deep brain stimulation studies might stem from different levels of neuronal activation by these various techniques.

## Potential cell-type-specificity in the ZI

In the present study, we evaluated the functional contribution of ZI-VGAT neurons to the modulation of pain-related behaviors. Our optogenetic-assisted circuit mapping experiments showed, however, that both VGAT-positive and VGAT-negative ZI neurons respond to optogenetic stimulation of CeA terminals in the ZI (*Figure 2C–D*). Consistent with these findings, previous studies have shown that CeA neurons, including somatostatin-positive cells, project to and inhibit parvalbumin-positive ZI

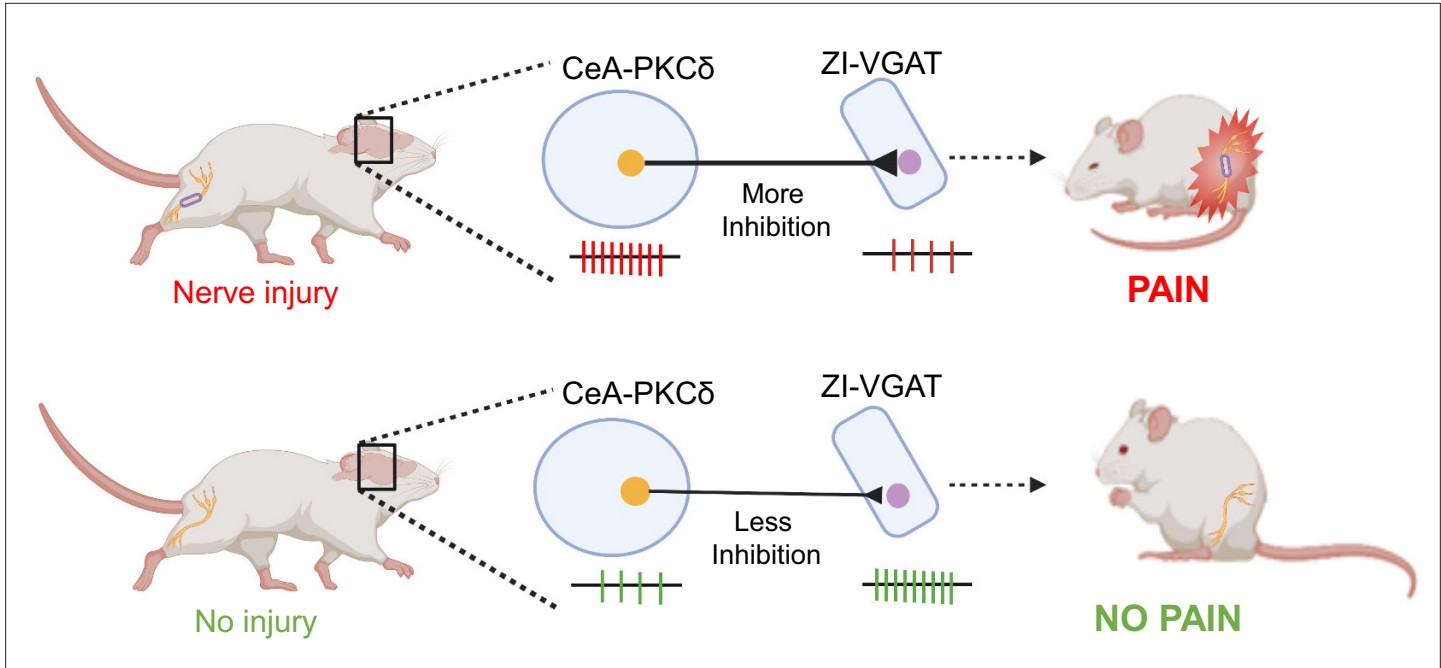

**Figure 9.** Proposed model for modulation of pain-related behaviors in the ZI. Schematic drawing showing activation of CeA-PKCδ neurons after nerve injury, which leads to inhibition of VGAT-positive ZI neurons (ZI-VGAT), subsequently promoting behavioral hypersensitivity.

neurons (*Zhou et al., 2018*). Whether somatostatin-positive CeA neurons also project to ZI-VGAT neurons and whether CeA-PKCδ neurons target a genetically or anatomically distinct subpopulation of ZI neurons remains to be determined. Future experiments to examine potential cell-type-specific pain-related plasticity of intrinsic excitability and synaptic transmission in this pathway will also be informative.

A recent study evaluated the function of parvalbumin-positive ZI neurons in the modulation of pain-related behaviors (*Wang et al., 2020*). Interestingly, this study demonstrated that parvalbumin-positive neurons in the ZI show an opposite electrophysiological phenotype and function in the modulation of pain-related behaviors than what we and others observe for the ZI-VGAT neurons. Thus, while the present and previous studies show that inhibition of ZI-VGAT neurons drives behavioral hypersensitivity, ablation or silencing of parvalbumin-positive ZI neurons was shown to be analgesic (*Wang et al., 2020*). Similarly, while we and others show that activation of ZI-VGAT neurons reverses injury-induced hypersensitivity, activation of parvalbumin-positive ZI neurons was shown to promote behavioral hypersensitivity (*Wang et al., 2020*). These combined results are particularly interesting given that parvalbumin-positive ZI neurons receive inputs from somatostatin-expressing CeA neurons (*Zhou et al., 2018*), which have been previously shown to be inhibited in the context of pain (*Wilson et al., 2019*).

Based on these combined findings, we predict that injury-induced inhibition of somatostatin-expressing neurons in the CeA disinhibits parvalbumin-positive ZI neurons, while injury-induced activation of CeA-PKCδ neurons inhibits a subpopulation of VGAT-positive ZI neurons, both resulting in behavioral hypersensitivity. Whether injury-induced inhibition of ZI-VGAT neurons (by CeA-PKCδ neurons) occurs simultaneously and under the same conditions as disinhibition of parvalbumin-positive ZI neurons (by somatostatin-expressing neurons in the CeA) remains to be determined.

# Materials and methods

**Key resources table**

| Reagent type (species) or resource | Designation | Source or reference | Identifiers | Additional information |
|---|---|---|---|---|
| Genetic reagent (*M. musculus*) | *PKCδ*-cre mice | GENSAT | founder line 011559-UCD | |
| Genetic reagent (*M. musculus*) | Ai9 mice | Jackson Laboratories | Stock number 007909 | |
| Genetic reagent (*M. musculus*) | Vesicular GABA transporter Cre mice | Jackson Laboratories | Stock number 016962 | |
| Genetic reagent (*M. musculus*) | C57BL/6NJ mice | Jackson Laboratories | Stock number 005304 | |
| Transfected construct (*M. musculus*) | pAAV8-hSyn-DIO-hM4D(Gi)-mCherry | Addgene; donated by Bryan Roth *Krashes et al., 2011* | Addgene:#44362-AAV8 | |
| Transfected construct (*M. musculus*) | pAAV8-hSyn-DIO-mCherry | Addgene; donated by Bryan Roth | Addgene:#50459-AAV8 | |
| Transfected construct (*M. musculus*) | pAAV8-hSyn-DIO-hM3D(Gq)-mCherry | Addgene; donated by Bryan Roth *Krashes et al., 2011* | Addgene:#44361-AAV8 | |
| Transfected construct (*M. musculus*) | AAV9-Syn-Flex-ChrimsonR-tdTomato | UNC; donated by Edward Boyden | Lot Number AV4384G | |
| Transfected construct (*M. musculus*) | rAAV2-hSyn-hChR2(H134R)-EYFP-WPRE-PA | UNC; donated by Karl Deisseroth | Lot Number AV6556C | |
| Transfected construct (*M. musculus*) | pAAV8-EF1a-double floxed-hChR2 (H134R)-mCherry-WPRE-HGHpA | Addgene; donated by Karl Deisseroth | Addgene:#20297-AAV8 | |
| Transfected construct (*M. musculus*) | AAV2-EF1a-DIO-eNpHR3.0-mCherry | UNC; donated by Karl Deisseroth | Lot Number AV4872B | |

*Continued on next page*

*Continued*

| Reagent type (species) or resource | Designation | Source or reference | Identifiers | Additional information |
|---|---|---|---|---|
| Transfected construct (*M. musculus*) | AAV2-EF1a-DIO mCherry | UNC; donated by Bryan Roth | Lot Number AV4735E | |
| Antibody (rat monoclonal) | rat anti-mCherr | Invitrogen | M11217 | 1:500 |
| Antibody (rabbit monoclonal) | rabbit anti-Phospho-c-Fos (Ser32) | Cell Signaling Technology | 5348 | 1:2000 |
| Antibody (mouse monoclonal) | mouse anti-PKCδ | BD Biosciences | 610397 | 1:1000 |
| Antibody (goat polyclonal) | goat anti-rat Cy3 | Invitrogen | A10522 | 1:250 |
| Antibody (goat polyclonal) | Alexa Fluor 647-conjugated goat anti-rabbit | Invitrogen | A21244 | 1:250 |
| Antibody (goat polyclonal) | Alexa Fluor 647-conjugated goat anti-mouse | Invitrogen | A21235 | 1:100 |
| Sequence-based reagent | Forward primer to genotype for the presence of cre-recombinase: TTAATCCATATTGGCAGAACGAAAACG | Transnetyx | Transnetyx.com | |
| Sequence-based reagent | Reverse primer to genotype for the presence of cre-recombinase: AGGCTAAGTGCCTTCTCTACA | Transnetyx | Transnetyx.com | |
| Peptide, recombinant protein | Alexa Fluor 647-conjugated cholera toxin subunit B | Invitrogen | C34778 | |

## Animals

Adult male mice (8–17–weeks old) were used for all experiments. Mice were housed in a vivarium with controlled humidity and temperature under a reversed 12 hr light/dark cycle (9 pm to 9 am light) with ad libitum access to food and water. All behavioral tests were performed during the dark period, between the hours of 10 am and 6 pm. Mice received 100 µl of saline intraperitoneal injection (i.p.) daily and were handled by the experimenter for one week before the start of behavioral and electro-physiological experiments following the cupping method as previously described (*Hurst and West, 2010*). Following surgeries, mice were housed in pairs and separated by a perforated Plexiglas divider. C57BL/6NJ mice (Jackson Laboratory; Stock number 005304) were bread as homozygous. Hetero-zygous male or female *Prkcd*-cre mice (GENSAT-founder line 011559-UCD), referred in this study as PKCδ-cre, were crossed with Ai9 mice (Jackson Laboratories; Stock number 007909). *Slc32a1*-ires-cre (Jackson Laboratories; Stock number 016962), referred in this study as vesicular GABA transporter (VGAT)-cre mice, were bred as homozygous pairs or crossed with Ai9 mice (Jackson Laboratories; Stock number 007909). Both the PKCδ-cre and VGAT-cre mouse lines have been previously validated and shown to express Cre recombinase selectively in PKCδ+and GABAergic neurons, respectively (*Vong et al., 2011*; *Wilson et al., 2019*). The presence of cre-recombinase in offspring was confirmed by genotyping using DNA extracted from tail biopsies. The primer sequences (Transnetyx) used for genotyping were TTAATCCATATTGGCAGAACGAAAACG (forward primer) and AGGCTAAGTGCCTTCTCTACA (reverse primer).

## Stereotaxic Injections in the CeA and ZI

Mice were initially anesthetized with 5% isoflurane in preparation for the stereotaxic surgery. After induction, mice were head-fixed on a stereotaxic frame (David Kopf Instruments) and 1.5–2% isoflu-rane at a flow rate of 0.5 L/min was used for the duration of surgery. A hand warmer was used for thermal maintenance during the procedure. Stereotaxic injections were performed using a 32-gauge needle on a 0.5 or 1.0 µl volume Hamilton Neuros syringe. All injections were performed at a flow rate of 0.1 µl/min and the syringe was left in place for an additional 5 min to allow for the diffusion of virus and to prevent backflow.

For evaluation of CeA-PKCδ terminal distribution within the brain, 0.3–0.6 µl of AAV8-hSyn-DIO-mCherry (Addgene viral prep #50459-AAV8) or AAV9-Syn-Flex-ChrimsonR-tdTomato (UNC GTC

Vector Core, AV4384G) was microinjected into the right CeA of PKCδ-cre mice. The coordinates were as follows: 1.4 mm posterior to bregma; 3.2 mm lateral to midline; 4.8 mm ventral to skull surface. To maximize transduction of cells within the whole rostral-caudal CeA, a subset of mice received a second stereotaxic injection of the AAV virus into the anterior right CeA (0.9 mm posterior to bregma; 3.2 mm lateral to midline; 4.8 mm ventral to skull surface). For retrograde labeling, 0.2 μl of Alexa Fluor 647-conjugated cholera toxin subunit B (Invitrogen, C34778) was stereotaxically injected into the right ZI of PKCδ-cre::Ai9 mice. The coordinates were as follows: 1.9 mm posterior to bregma; 1.4 mm lateral to midline; 4.75 mm ventral to skull surface. For optogenetic-assisted circuit mapping experiments, 0.3 μl of AAV2-hSyn-hChR2(H134R)-EYFP (UNC GTC Vector Core, AV6556C) was micro-injected into the right CeA of VGAT-cre::Ai9 mice. The coordinates were as follows:1.25 mm posterior to bregma; 3.0 mm lateral to midline; 4.5 mm ventral to skull surface. For chemogenetic experiments, 0.15 μl of AAV8-hSyn-DIO-hM4D(Gi)-mCherry (Addgene viral prep #44362-AAV8;), AAV8-hSyn-DIO-hM3D(Gq)-mCherry (Addgene viral prep #44361-AAV8; *Krashes et al., 2011*) or AAV8-hSyn-DIO-mCherry (Addgene viral prep #50459-AAV8) were microinjected into the right ZI of VGAT-cre mice. The coordinates were as follows: 1.7 mm posterior to bregma, 1.2 mm lateral to midline; 4.7 mm ventral to skull surface. A minimum of 2 weeks was given between brain injections and chemogenetic behavior testing to allow for efficient viral-mediated transduction. For anatomical experiments, we waited a minimum of 4 weeks between the brain injections and the experiments.

## In vivo optogenetics

For optogenetic manipulation of CeA-PKCδ terminals in the ZI, AAV8-EF1a-DIO-hChR2(H134R)-mCherry-WPRE-HGHpA, AAV2-EF1a-DIO-eNpHR3.0-mCherry, or AAV2-EF1a-DIO mCherry control virus was infused into the right CeA of PKCδ-cre mice. Two 0.6 μl injections in the anterior and posterior CeA were performed to maximize transduction of cells within the whole rostral-caudal CeA using the following stereotaxic coordinates: 0.9 mm and 1.40 mm posterior to bregma; 3.2 mm lateral to midline; 4.8 mm ventral to skull surface. Immediately after viral injection, mice were implanted with optical probes (Thorlabs CFMLC22L05; core diameter 200 μM; 0.22 NA; 6 mm length) over the right ZI using the following coordinates: 1.70 mm posterior from bregma; 0.90 mm lateral to midline; 4.8 mm ventral to skull surface. Ferrules were secured to the skull with meta bond (Parkell Prod). Mice were allowed to recover from the surgery for 3 weeks before behavior testing. Optogenetic stimulation was performed using blue (473 nm; intensity 0.1–10 mW; 10 Hz frequency) or yellow (561 nm; intensity 0.04–10 mW; 10 Hz frequency) lasers. For optogenetic experiments, we waited a minimum of 3 weeks between the brain injections and the experiments.

## Sciatic cuff implantation

Sciatic cuff implantation surgeries were performed 1 week after the brain surgeries as previously described (*Benbouzid et al., 2008*). Briefly, mice were anesthetized with 2% isoflurane at a flow rate of 0.5 L/min. An incision of about 1 cm long was made in the proximal one third of the lateral left thigh. The sciatic nerve was externalized and stretched using forceps. For the cuff-implanted group, a polyethylene tubing PE20 (2 mm-long, 0.38 mm ID / 1.09 mm OD; Daigger Scientific) was slid onto the sciatic nerve and was then placed back in its location. Similarly, for the comparative sham group, mice went through the same process of sciatic nerve exposure and stretching but no tubing was implanted. After the procedure was complete, the skin above the thigh was closed with wound clips. The mice were subjected to at least a one-week recovery period before performing the behavior tests.

## Nociceptive behaviors

Behavioral experiments were performed under red light, during the dark phase, and the experimenter was blind to treatment conditions. Mice were randomized into control and experimental groups. Individual cohorts were counterbalanced to include mice from all experimental groups. The time-line for the behavioral chemogenetic experiments relative to AAV brain injections were as follows: Acetone test 14–15 days; Hargreaves test: 16–17 days; von Frey test: 20–22 days; Randall-Selitto test: 23–25 days. Each test was performed on 2 consecutive days. On each testing day, baseline (pre-injection) measurements were taken. Saline or Clozapine-N-oxide (CNO, Enzo Life Sciences, Farming-dale, NY) was injected i.p. (10 mg/kg body weight for hM4Di and 5 mg/kg for hM3Dq experiments) and a second measurement (post-injection) was taken 45 min to 1 hour after the i.p. injection. Mice

were randomly assigned to saline or CNO on the first day of each test and were tested on the opposite treatment on the second day of the same test. Data is represented as an average score of 5 stimulations from each hindpaw before, 1 hour and 1 day after drug treatment (CNO or vehicle).

The timeline for optogenetic experiments relative to AAV brain injections and optic probe implantation were as follows: Acetone and von Frey test 21–22 days; Hargreaves 23–24 days; Randall-Selitto test 25–26 days. Data is represented as an average score of 5 stimulations from each hindpaw before and during and after laser stimulation.

At the end of each experiment, mice were transcardially perfused with 4% paraformaldehyde solution in 0.1 M Phosphate Buffer (PFA/PB), pH 7.4 and the brains were stained for mCherry as described below to verify injection sites and placement of the optical probe tip. Anatomical limits of the ZI and CeA were identified using a mouse brain atlas (*Paxinos et al., 2001*). Drawings of the virus spread as a function of the rostral-caudal level were performed for each injected mouse and only mice that had virus injection restricted to the ZI were used for behavioral chemogenetic analyses. Only mice with correct optic probe placement in ZI and virus injection restricted to CeA were used for behavioral optogenetic analyses.

## Acetone test

Cold hypersensitivity was assessed by the acetone evaporation test as previously described (*Choi et al., 1994*). Briefly, mice were habituated for at least 2 hr in individual ventilated opaque white Plexiglas testing chambers (11x11 x 13 cm) on an elevated platform with a floor made of wire mesh. An acetone drop was formed at the top of a 1 mL syringe and gently touched to the center of the plantar surface of the hindpaw ipsilateral or contralateral to sciatic nerve surgery. Nociceptive responses to the acetone drop were evaluated for 60 s using a modified 0–2-point system developed by *Colburn et al., 2007*. According to this scoring system, 0=rapid, transient lifting, licking, or shaking of the hindpaw, which subsides immediately; 1=lifting, licking, and/or shaking of the hindpaw, which continues beyond the initial application, but subsides within 5 s; 2=protracted, repeated lifting, licking, and/or shaking of the hindpaw. Five trials were performed with ~5 min between-trial intervals.

## Hargreaves test

To evaluate heat hypersensitivity, we used a modified version of the Hargreaves Method (*Hargreaves et al., 1988*) as previously described (*Wilson et al., 2019*). On the experiment day, mice were habituated prior to testing for at least 1 hr in individual ventilated opaque white plexiglass testing chambers (11x11 x 13 cm) placed on an elevated glass floor maintained at 30 °C. Following habituation, a noxious radiant heat beam was applied through the glass floor (IITC Life Sciences, Woodland Hills, CA) to the center of the plantar surface of the hindpaw (ipsilateral or contralateral to sciatic nerve surgery), until the mouse showed a withdrawal response. A cutoff of 15 s latency and 25 active intensity was used in each trial to prevent skin lesions. At least 3 min were allowed between consecutive trials. The average of five trials was calculated and used as the threshold for each hindpaw.

## Von Frey

Mechanical hypersensitivity was assessed as the paw withdrawal threshold in response to von Frey filaments (North Coast Medical, Inc San Jose, CA), as previously described (*Wilson et al., 2019*). On each testing day, mice were placed individually in ventilated opaque white Plexiglas testing chambers (11x11 x 13 cm) on an elevated mesh platform at least 2 hr before application of stimulus. A mesh floor allowed full access to the paws from below. After the acclimation period, each von Frey filament was applied to the center of the plantar surface of the hindpaw (ipsilateral or contralateral to sciatic nerve surgery) for 2–3 s, with enough force to cause slight bending against the paw. This procedure continued for a total of five measurements. The smallest filament that evoked a paw withdrawal response in at least three of five measurements was taken as the mechanical threshold for that trial. The average of five trials was calculated and used as the threshold value per hindpaw.

## Randall-Selitto test

The Randall-Selitto test was performed to assess the response thresholds to mechanical pressure stimulation (pinch) of the hindpaws *Randall and Selitto, 1957* in lightly anesthetized animals. Briefly, mice were anesthetized in 5% isoflurane in an induction chamber. Subsequently, animals were kept under

light anesthesia with 0.5–1% isoflurane at a flow rate of 0.5 L/min. A maximal cut-off of 200 g force was delivered to the plantar surface to prevent tissue damage. Five trials per animals were recorded and the average was calculated.

## Slice electrophysiology

Acute coronal ZI and CeA slices were prepared from brains of VGAT-cre, VGAT-cre::Ai9, or PKCδ-cre mice (9–18 week-old) 2–8 weeks after stereotaxic injection of AAV8-hSyn-DIO-hM4Di-mCherry into the ZI or injection of AAV2-hSyn-hChR2(H134R)-EYFP, AAV8-EF1a-DIO-hChR2(H134R)-mCherry-WPRE-HGHpA or AAV2-EF1a-DIO-eNpHR3.0-mCherry into the CeA. Briefly, mice were deeply anesthetized with 1.25% Avertin anesthesia (2,2,2-tribromoethanol and tert-amyl alcohol in 0.9% NaCl; 0.025 ml/g body weight). For ZI slices, mice were sacrificed by cervical dislocation and decapitated. For CeA slices, mice were transcardially perfused with ice-cold cutting solution containing (in mM): 110.0 choline chloride, 25.0 NaHCO$_3$,1.25 NaH$_2$PO$_4$, 2.5 KCl, 0.5 CaCl$_2$, 7.2 MgCl$_2$, 25 D-glucose,12.7 L-ascorbic acid, 3.1 pyruvic acid, and saturated with 95% O$_2$-5% CO$_2$. After cervical dislocation or perfusion, brains were rapidly removed and placed in ice-cold cutting solution. Coronal slices (250–300 μm) containing the ZI or the CeA were cut on a Leica VT1200 S vibrating blade microtome (Leica Microsystems Inc, Buffalo Grove, IL, USA) and incubated in a holding chamber with oxygenated artificial cerebral spinal fluid (ACSF) containing (in mM): 125 NaCl, 2.5 KCl, 1.25 NaH$_2$PO$_4$, 25 NaHCO$_3$, 2.0 CaCl$_2$,1.0 MgCl$_2$, 25 D-glucose (~310 mOsm) saturated with 95% O$_2$- 5% CO$_2$, at 33 ° C for 30 min, then moved to room temperature for a minimum of 20 min before transfer to the recording chamber.

All recordings were performed at 33 ± 1° C and using potassium methylsulfate-based internal solution in mM: 120 KMeSO$_4$, 20 KCl, 10 HEPES, 0.2 EGTA, 8 NaCl, 4 Mg-ATP, 0.3 Tris-GTP, 14 Phospho-creatine, pH 7.3 with KOH (~300 mosmol-1) unless otherwise stated. A recording chamber heater and an in-line solution heater (Warner Instruments) were used to control and monitor the bath temperature throughout the experiment. Cells were visually identified using an upright microscope (Nikon Eclipse FN1) equipped with differential interference contrast optics with infrared illumination and fluorescent microscopy. Transduced cells were visually identified based on their expression of the mCherry or EYFP fluorophore. Current clamp signals were acquired at 100 kHz and filtered at 10 kHz. Voltage clamp signals were acquired at 10 kHz and filtered at 2 kHz.

To validate the effects of CNO on hM4Di-transduced cells in the ZI, current-clamp recordings were used to assess changes in excitability. Spontaneously active cells were injected with hyperpolarizing current (−10 to −50 pA) to bring their membrane potentials to between −80 and −70 mV. A 500ms depolarizing current (10–120 pA) was injected to elicit between 2 and 5 action potentials. The current injection repeated every 15 s until the cell fired stably and consistently. Following this stabilization, 10 additional recordings were acquired before bath application of either 10 μM CNO or vehicle in ACSF. Recordings of the same current injection were continued every 15 s for approximately 5 min, until the cell fired consistently and stably. Ten additional recordings were performed. The number of action potentials elicited during each depolarizing current injection were used to assess excitability. Values for before and after CNO or vehicle application were averaged across five traces and compared.

To validate the optogenetics effects, current- and voltage-clamp recordings were performed on CeA neurons transduced with ChR2 or NpHR of PKCδ-cre mice previously injected with ChR2-mCherry or NpHR-mCherry into the CeA. Non-transduced neurons within the CeA region containing transduced cells were used as controls. For ChR2 experiments, voltage and current responses of neurons in response to a 1 s blue LED ($\lambda$ =470 nm, Mightex) stimulation of 10 Hz and 5ms duration were recorded. Neurons were held between −60 mV and −50 mV with a depolarizing current (20–90 pA) for current-clamp recordings and at −70 mV for voltage-clamp recordings. Light-evoked inward currents were used to assess responses in voltage-clamp experiments. The number of light-evoked action potentials within the 1 s stimulation in current-clamp experiments was used to assess spike fidelity in each neuron.

For NpHR experiments, neurons were held at −70 mV and prolonged (2 s) depolarizing current injections were used to elicit repetitive firing. The amplitude of the current injections (70–700 pA) was adjusted per neuron to elicit between 5 and 10 action potentials within the 400ms prior to light stimulation. A 400ms duration yellow light ($\lambda$ =590 nm, Mightex) stimulation was used to assess NpHR-mediated inhibition. The number of action potentials elicited 400ms before and during the 400ms yellow light stimulation was quantified in each cell.

## Optogenetically assisted circuit mapping

For the optogenetically assisted circuit mapping experiments, recordings were performed using a cesium gluconate-based internal solution containing (in mM): 120 cesium gluconate, 6 NaCl, 10 HEPES, 12 phosphocreatine, 5 EGTA, 1 $CaCl_2$, 2 $MgCl_2$, 2 ATP, and 0.5 GTP, pH 7.4 adjusted with CsOH (~290 mOsm). Whole-cell voltage-clamp recordings were obtained at 33 ± 1° C from visually identified tdTomato-expressing and non-expressing ZI neurons using differential interference contrast optics with infrared illumination and fluorescence microscopy. Optically evoked inhibitory postsynaptic currents (oIPSCs) of VGAT + and VGAT- neurons were recorded at a holding potential of 0 mV in the presence of TTX (1 μM) and 4-AP (100 μM). Blue LED light ($\lambda$ =470 nm, 10–12 mW, Mightex) paired pulses of 0.5–10ms duration with an interval of 200ms between pulses were delivered to drive paired synaptic responses. Paired pulse ratios (PPR) were determined by the ratio of the amplitude of the peak evoked by the second pulse divided by the amplitude of the peak evoked by the first pulse. Signals were acquired at 100 kHz and filtered at 10 kHz. The injection site in the CeA was verified in acute brain slices prior to recording and only mice that had virus injection restricted to the CeA and robust terminal labeling in the ZI were used for circuit mapping electrophysiological experiments.

To validate a functional inhibitory projection between the CeA and ZI, acute coronal slices were prepared from male C57BL/6 J mice (12–18 weeks) previously injected with AAV2-hsyn-hChR2(H134R)-EYFP using the slice preparation protocol mentioned above. Post-synaptic currents in response to blue light ($\lambda$ =470 nm, 10–12 mW, Mightex) stimulation of 5ms duration were recorded at a holding potential of 0 mV or –70 mV in ACSF. Parallel recordings were obtained before and 3 min after bath exchange to ACSF containing 10 μM Bicuculline (Sigma). Peak amplitudes of light-evoked responses were averaged across 10 recording sweeps. A 10 mM stock solution of Bicuculline was made in water, stored at –20° C and dissolved in ACSF on the day of the experiment.

## Immunohistochemistry

At the end of each experiment, mice were deeply anesthetized with 1.25% Avertin anesthesia (2,2,2-tribromoethanol and tert-amyl alcohol in 0.9% NaCl; 0.025 ml/g body weight) i.p., then perfused transcardially with 0.9% NaCl (37 °C), followed by 100 mL of ice-cold 4% paraformaldehyde in phosphate buffer solution (PFA/PB). The brain was dissected and post fixed in 4% PFA/PB overnight at 4 °C. After cryoprotection in 30% sucrose/PB for 48 hr, coronal sections (30–45 μm) were obtained using a freezing sliding microtome and stored in 0.1 M Phosphate Buffered Saline (PBS), pH 7.4 containing 0.01% sodium azide (Sigma) at 4 °C until immunostaining. Sections were rinsed in PBS, incubated in 0.1% Triton X-100 in PBS for 10 min at room temperature and blocked in 5% normal goat serum (NGS) (Vector Labs, Burlingame, CA) with 0.1% Triton X-100, 0.05% Tween-20 and 1% bovine serum albumin (BSA) for 30 min at room temperature. Sections were then incubated for 72 hr at 4 °C in mouse anti-PKCδ (1:1000, BD Biosciences, 610397), rabbit anti-Phospho-c-Fos (1:2000, Cell Signaling Technology, 5348) or rat anti-mCherry (1:500, Invitrogen, M11217) in 1.5% NGS blocking solution with 0.1% Triton X-100, 0.05% Tween-20 and 1% BSA. Sections were then rinsed in PBS and incubated in Alexa Fluor 647-conjugated goat anti-mouse (1:100, Invitrogen, A21235), Alexa Fluor 647-conjugated goat anti-rabbit (1:250, Invitrogen, A21244), or goat anti-rat Cy3 (1:250, Invitrogen, A10522) secondary antibodies in 1.5% NGS blocking solution with 0.1% Triton X-100, 0.05% Tween 20 and 1% BSA, protected from light, for 2 hr at room temperature. Sections were then rinsed in PBS, mounted on positively charged glass slides, air-dried and coverslips were placed using Fluoromount-G (Southern Biotech).

For the c-Fos experiments, VGAT-cre mice received CNO (5 mg/kg) or saline injections (i.p.) 2 weeks post virus injection into the ZI. Mice were housed in their home cages for 1 hr prior to transcardial perfusion, brain dissection and tissue processing. For mapping of axonal terminals from CeA-PKCδ neurons, PKCδ-cre mice injected with AAV9-Syn-Flex-ChrimsonR-tdTomato or AAV8-hSyn-DIO-mCherry were transcardially perfused at least 4 weeks after the brain injections. For the mCherry-injected brains, 30 μm coronal sections from the entire brain were collected and immunostained for mCherry as described above.

## Imaging and analysis

For confocal studies, images were acquired using a Nikon A1R laser scanning confocal microscope. ×2 (for low magnification), ×20 (for high magnification) or ×40 (oil-immersion for higher magnification)

objectives were used. The experimental conditions for image collection including laser intensity, gain, and pinhole size were identical for experiments. Multiple channels (GFP, RFP and CY5) were used for sequential image acquisition where Z stacks data collection was done at 0.9 mm. Following acquisition, images were consolidated using NIS Elements software with automatic stitching of subsequent images, and conversion of stacks into maximum intensity z-projections. Quantitative analysis of CeA and ZI imaging data was performed between bregma –0.82 and –1.94 and bregma –1.06 and –2.54, for CeA and ZI, respectively. Anatomical limits of each region were identified based on the mouse brain atlas (*Paxinos et al., 2001*). Number of positive cells were quantified manually for each channel using NIS Elements software using one section per rostral-caudal level for each mouse. Co-labeled cells were identified by NIS Elements software automatically and were further visually corroborated by an experimenter.

For mapping of axonal terminals from CeA-PKCδ neurons, coronal slices from the entire brain of PKCδ-cre mice injected with AAV9-Syn-Flex-ChrimsonR-tdTomato or AAV8-hSyn-DIO-mCherry, collected and immunostained 120 µm apart from each other, were visually inspected for the presence of fluorescent axonal terminals using a 20 X objective in a Nikon A1R laser scanning confocal microscope. Only mice that had injections restricted to the CeA were used for anatomical experiments. Classic morphological criteria, defined as the presence of varicosities and the thickness and organization pattern of the signal, was used to distinguish labeled terminals (very thin fibers with numerous ramifications and varicosities) from fibers of passage (thicker fibers without ramifications and varicosities) as previously described (*Bernard et al., 1993*). Low and high magnification images of all the brain sections containing terminals were acquired and the anatomical localization of the terminals was then determined using a Mouse Brain Atlas (*Paxinos et al., 2001*). Representative images of terminals were collected using a 40 X oil-immersion objective.

Semi-quantitative analysis of the areas containing axonal terminals was done based on the density of terminals observed and are reported as sparse (+), moderate (++) and dense (+++) in the defined area.

A similar analysis was also performed with the brain of experiment 265945645 of the Mouse Brain Connectivity Atlas of the Allen Brain Institute (http://connectivity.brain-map.org/). This brain was identified using the Source Search tool on the Mouse Brain Connectivity Atlas website and filtering for the CeA as the brain region and *PKCδ*-GluCla-CFP-IRES-Cre as the mouse line of interest. The tracer injected is described as EGFP; the stereotaxic coordinates for the injection were AP –1.82 mm, ML –2.65 mm, DV –4.25 mm and the injection volume was 0.02 mm$^3$.

Quantitative analysis of axonal terminal densities was performed by measuring the percentage of area containing moderate immunofluorescent terminals within the ZI. Terminal labeling was automatically detected using the NIS Elements software and a pre-defined signal intensity threshold followed by visual corroboration by an experimenter.

## Data and statistical analyses

The sample sizes used in each experiment were based on the standards set forth by the field. Five mice were used as biological replicates for anatomical tracing experiments. At least two mice were used as biological replicates for electrophysiology and histological experiments and at least 4 mice were used as biological replicates for behavioral experiments. Data are presented as mean ± SEM. Statistical analyses was conducted using GraphPad Prism (v8.0). Unpaired/paired two-tailed t-test, Wilcoxon two-tailed matched pair signed rank test, one-way analyses of variance (ANOVA) followed by Tukey's multiple comparison test and two-way repeated measures ANOVA followed by Dunnett's multiple comparison test were used. The significance level was set at <0.05. Sample sizes and p values are indicated in figure legends.

## Acknowledgements

We thank Drs. Yavin Shaham and Maria Luisa Torruella-Suarez for comments on this manuscript and Simón Arango, Adela Francis-Malavé and Jeitzel Torres-Rodriguez for assistance in histological and anatomical experiments. pAAV-hSyn-DIO-hM3D(Gq)-mCherry (Addgene viral prep # 44361-AAV8), pAAV-hSyn-DIO-hM4D(Gi)-mCherry (Addgene viral prep # 44362-AAV8), pAAV8-hSyn-DIO-mCherry (Addgene viral prep # 50459-AAV8), and AAV2-EF1a-DIO mCherry (UNC GTC vector core) were a gift from Bryan Roth. pAAV8-EF1a-double floxed-hChR2 (H134R)-mCherry-WPRE-HGHpA

(Addgene viral prep # 20297-AAV8) was a gift from Karl Deisseroth. AAV9-Syn-Flex-ChrimsonR-tdTomato was provided by Vector Core at the University of North Carolina, with a material transfer agreement with Edward Boyden. AAV2-EF1a-DIO-eNpHR3.0-mCherry and AAV2-hsyn-hChR2(H134R)-EYFP were provided by the Vector Core at the University of North Carolina, with a material transfer agreement with Karl Deisseroth (Stanford University). Cartoons in figures were created with BioRender.com.

## Additional information

### Funding

| Funder | Grant reference number | Author |
|--------|------------------------|--------|
| National Center for Complementary and Integrative Health | Intramural Research Program | Yarimar Carrasquillo |
| National Institute of Mental Health | Intramural Research Program | Mario A Penzo |

The funders had no role in study design, data collection and interpretation, or the decision to submit the work for publication.

### Author contributions

Sudhuman Singh, Conceptualization, Data curation, Formal analysis, Validation, Investigation, Visualization, Methodology, Writing - original draft, Writing - review and editing; Torri D Wilson, Data curation, Validation, Investigation, Methodology, Writing - review and editing; Spring Valdivia, Conceptualization, Data curation, Formal analysis, Validation, Investigation, Visualization, Methodology, Writing - review and editing; Barbara Benowitz, Sarah Chaudhry, Data curation, Formal analysis, Validation, Investigation, Visualization, Methodology, Writing - original draft, Writing - review and editing; Jun Ma, Investigation, Methodology; Anisha P Adke, Data curation, Formal analysis, Validation, Investigation, Methodology, Writing - review and editing; Omar Soler-Cedeño, Data curation, Formal analysis, Validation, Investigation, Methodology; Daniela Velasquez, Conceptualization, Data curation, Investigation, Methodology; Mario A Penzo, Supervision, Funding acquisition, Methodology, Writing - review and editing; Yarimar Carrasquillo, Conceptualization, Data curation, Formal analysis, Supervision, Funding acquisition, Validation, Investigation, Visualization, Methodology, Writing - original draft, Project administration, Writing - review and editing

### Author ORCIDs

Sudhuman Singh  http://orcid.org/0000-0002-9999-9189
Spring Valdivia  http://orcid.org/0000-0002-4282-2390
Barbara Benowitz  http://orcid.org/0000-0003-2627-1709
Sarah Chaudhry  http://orcid.org/0000-0002-2152-6947
Jun Ma  http://orcid.org/0000-0002-6404-6695
Anisha P Adke  http://orcid.org/0000-0001-5062-3319
Omar Soler-Cedeño  http://orcid.org/0000-0001-5897-4156
Mario A Penzo  http://orcid.org/0000-0002-5368-1802
Yarimar Carrasquillo  http://orcid.org/0000-0002-0702-4975

### Ethics

All experiments were approved by the Animal Care and Use Committee of the National Institute of Neurological Disorders and Stroke and the National Institute on Deafness and other Communication Disorders with the guidelines set by the National Institutes of Health (ASP1397).

### Decision letter and Author response

Decision letter https://doi.org/10.7554/eLife.68760.sa1
Author response https://doi.org/10.7554/eLife.68760.sa2

# Additional files

## Supplementary files
• Transparent reporting form

## Data availability
All data generated during this study are included in the manuscript and supporting files. Source data files have been provided for Figures 1-8 (including supplemental figures).

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
