## [Editor Report]

This manuscript from Singh and colleagues investigates neural connections between the central amygdala and the zona incerta, two subcortical brain regions previously implicated in pain, and further describes the role of the zona incerta to preclinical pain-related behavior in mice. This study employed anatomical tracing, electrophysiology, optogenetics, chemogenetics, and behavioral assays in various pain modalities to link the zona incerta to pain modulation by providing new evidence for a direct inhibitory connection from the central amygdala to the zona incerta that could explain neuropathic pain hypersensitivity. This study is detailed anatomically, electrophysiologically, and behaviorally, and the inclusion of optogenetic studies has enhanced the conclusions. While there are still some confirmatory conclusions from prior work, the detail and execution of this study enhance the field.

---

## [Decision Letter]

**Decision letter after peer review:**

Thank you for submitting your article "Bidirectional modulation of pain-related behaviors in the zona incerta" for consideration by *eLife*. Your article has been reviewed by 3 peer reviewers, and the evaluation has been overseen by a Reviewing Editor and Kate Wassum as the Senior Editor. The following individual involved in review of your submission has agreed to reveal their identity: Volker Neugebauer (Reviewer #3).

Essential revisions:

Following review, all 3 reviewers agree that there is potential interest in this manuscript and relevance for the field, but all were in agreement that some core data was missing to link the PKCδ neurons in the CeA projecting to the ZI as being the explicit circuit of interest. The electrophysiology experiments did not discriminate which CeA inputs were activated with optogenetics to modulate GABA release in ZI, and therefore there is not sufficient evidence to claim that PKCδ neurons in the CeA are explicitly driving changes in the ZI. More importantly, all of the behavioral studies were performed only via manipulation of GABA neurons in the ZI itself, with no experiments actually verifying that the PKCδ neuronal input from the CeA was involved in any of the changes in pain. It was felt that the primary novelty of this study related to the identification that PKCδ neurons in the CeA projecting to the ZI were relevant for the modulation of pain, and accordingly, it was unanimously agreed that for this paper to be considered acceptable for publication additional studies would have to be performed for the authors to substantiate and support the conclusions of this this paper.

To this extent, it was agreed that it was necessary for the authors to perform an additional behavioral study where they explicitly manipulate CeA-PKCδ neuronal inputs to the ZI using either opto- or chemo-genetic approaches. All of the reviewers agreed that these manipulations would have to be performed on CeA PKCδ neuron axon terminals in the ZI, so if it was an optogenetic approach the fiber would have to be placed over the ZI and if it was a chemogenetic approach the DREADD ligand would have to be locally infused directly into the ZI, to influence the inputs from CeA PKCδ neurons in the ZI following infection of CeA PKCδ neurons with the appropriate construct. In the absence of this additional study, the reviewers all agreed the paper could not be considered further.

Second, it was also agreed upon that, since in its current form the authors do not demonstrate that it is CeA PKCδ cells that modulate ZI neurons, they have the choice to remove any such statement (functional monosynaptic connections from CeA PKCδ to ZI) or perform additional electrophysiology experiments to support this conclusion. DIO-ChR2 virus could be injected into the CeA of PKCδ-cre mice and optically evoked IPSCs could be recorded from ZI neurons. The addition of these experiments was not viewed as being essential to be considered further, but as stated, if the authors choose to not add these experiments they have to substantially modify the conclusions of this paper to remove any direct reference suggesting they have demonstrated functional monosynaptic connections within this circuit.

In addition to these essential revisions raised by reviewers, there were also additional comments that required attention:

– The addition of comparable PPR recording from sciatic cuff animals or another injury would be useful.

– It would be interesting to compare IPSCs between sham control and neuropathic pain model to see if the (inhibitory) input from CeA-PKCδ neurons was increased in the pain model as hypothesized by the authors.

– It is not clear why a red-shifted channel rhodopsin (CrimsonR-tdTomato) was used as an anterograde tracer. Since optogenetics was not used in these experiments, control AAV that expressed only tdTomato would be appropriate. The rationale is not clear.

– It is also difficult to appreciate CrimsonR-tdTomato transduced cells (high magnification images in Figure 1B).

– "ChrimsonR" should be changed to "ChrimsonR-tdTomato" throughout, because tdTomato is the fluorophore we see in the images.

– Figure 1- Supplemental figure 1(C) could benefit from higher magnification/resolution images of PKCδ terminals in the ZI. It looks like some terminals were labeled outside of ZI. Quantification of terminals in the ZI would be important because the focus of this study is on the CeA-PKCδ to ZI projections.

– Controls are needed for electrophysiology experiments to show blockade of optogenetically evoked IPSCs with bicuculine and a negative control at -70mV.

– Behavioral results are convincing but would benefit from more rigorous presentation. It is pretty much standard now to show individual data points to appreciate variability of the data. SEMs of behavioral data are amazingly small.

– Since modality specific behavioral effects were found, it would also be important to include other behavioral outcome measures (e.g., non-evoked).

– Consistent use of terminology for the viral vectors is suggested.

---

## [Author Response]

Essential revisions:Following review, all 3 reviewers agree that there is potential interest in this manuscript and relevance for the field, but all were in agreement that some core data was missing to link the PKCδ neurons in the CeA projecting to the ZI as being the explicit circuit of interest. The electrophysiology experiments did not discriminate which CeA inputs were activated with optogenetics to modulate GABA release in ZI, and therefore there is not sufficient evidence to claim that PKCδ neurons in the CeA are explicitly driving changes in the ZI. More importantly, all of the behavioral studies were performed only via manipulation of GABA neurons in the ZI itself, with no experiments actually verifying that the PKCδ neuronal input from the CeA was involved in any of the changes in pain. It was felt that the primary novelty of this study related to the identification that PKCδ neurons in the CeA projecting to the ZI were relevant for the modulation of pain, and accordingly, it was unanimously agreed that for this paper to be considered acceptable for publication additional studies would have to be performed for the authors to substantiate and support the conclusions of this this paper.

We agree that a direct link between behavioral hypersensitivity and the anatomical findings showing CeA-PKCδ projections to ZI was missing. We have addressed this important concern in the revised version as described below. We chose to change the title of the paper to convey this important change in the take-home message of the manuscript. The new title is “An inhibitory circuit from central amygdala to zona incerta drives pain-related behaviors in mice”.

To this extent, it was agreed that it was necessary for the authors to perform an additional behavioral study where they explicitly manipulate CeA-PKCδ neuronal inputs to the ZI using either opto- or chemo-genetic approaches. All of the reviewers agreed that these manipulations would have to be performed on CeA PKCδ neuron axon terminals in the ZI, so if it was an optogenetic approach the fiber would have to be placed over the ZI and if it was a chemogenetic approach the DREADD ligand would have to be locally infused directly into the ZI, to influence the inputs from CeA PKCδ neurons in the ZI following infection of CeA PKCδ neurons with the appropriate construct. In the absence of this additional study, the reviewers all agreed the paper could not be considered further.

We thank the reviewers for raising this important point. We agree that behavioral experiments to evaluate the function of CeA-PKCδ neuron projections to ZI is crucial to test our hypothesis. To address this concern, we performed in-vivo optogenetic manipulation of CeA-PKCδ terminals in ZI. Consistent with our hypothesis, we found that optogenetic inhibition of CeA-PKCδ terminals in ZI reverses peripheral hyperalgesia in the injured paw. In contrast, optogenetic excitation of CeA-PKCδ terminals in ZI induces peripheral hypersensitivity in the uninjured hind paws in mice. We also performed ex-vivo slice electrophysiology experiments to validate optical inhibition and activation of CeA-PKCδ neurons transduced with NpHR or ChR2 by yellow or blue light, respectively. Together, these results demonstrate the function of the CeA-PKCδ to ZI pathway in the modulation of nociceptive behavior in mice. The results from these experiments are included in Figures 5-8, pages 15-18, and lines 340-411 of the revised version.

Second, it was also agreed upon that, since in its current form the authors do not demonstrate that it is CeA PKCδ cells that modulate ZI neurons, they have the choice to remove any such statement (functional monosynaptic connections from CeA PKCδ to ZI) or perform additional electrophysiology experiments to support this conclusion. DIO-ChR2 virus could be injected into the CeA of PKCδ-cre mice and optically evoked IPSCs could be recorded from ZI neurons. The addition of these experiments was not viewed as being essential to be considered further, but as stated, if the authors choose to not add these experiments they have to substantially modify the conclusions of this paper to remove any direct reference suggesting they have demonstrated functional monosynaptic connections within this circuit.

We agree that this is an important issue and attempted to address it as outlined below. We recorded from 16 neurons of 5 PKCδ-cre mice injected with cre-dependent ChR2 into the CeA but observed no postsynaptic responses. High transduction ChR2 efficiency was seen in the CeA and ChR2-expressing terminals were visible in the ZI in slices from all 5 mice. Given that these recordings were performed using a Cs-gluconate-based internal solution and no synaptic blockers in the bath, we reasoned that using a high chloride internal solution (CsCl) in the presence of excitatory synaptic blockers (NBQX and AP5) in the bath would increase the likelihood of observing a response. Thus, we recorded from 14 additional neurons from 3 additional PKCδ-cre mice with cre-dependent ChR2 injections in the CeA but again observed no response. These 3 mice also had robust ChR2 transduction in the CeA and visible ChR2expressing terminals in the ZI. We concluded that the moderate terminal labeling from CeA-PKCδ neurons in the ZI decreases the probability of patching a neuron that receives CeA-PKCδ inputs, making these experiments challenging and the results inconclusive. Based on previous studies showing that ZI neurons branch widely along the length of the ZI

(https://doi.org/10.1152/jn.00423.2006; https://doi.org/10.1523/JNEUROSCI.3768-06.2007), it is also possible that the ZI neurons receiving input from CeA-PKCδ neurons are not the neurons immediately abutting the highest terminal density. Axodendritic inputs of this kind would again make it difficult to target the appropriate neurons given the size of the ZI. Our CTB results (Figure 2), however, clearly confirmed an anatomical connection between CeA-PKCδ neurons and the ZI.

The new optogenetic experiments showing behavioral effects after manipulating CeA-PKCδ terminals in the ZI further confirm the functionality of this pathway in mediating pain-related behaviors. Whether the behavioral effects observed are mediated monosynaptically or polysynaptically remains to be determined. Thus, we toned down our conclusions related to a monosynaptic connection throughout the revised version of the manuscript.

In addition to these essential revisions raised by reviewers, there were also additional comments that required attention:– The addition of comparable PPR recording from sciatic cuff animals or another injury would be useful.– It would be interesting to compare IPSCs between sham control and neuropathic pain model to see if the (inhibitory) input from CeA-PKCδ neurons was increased in the pain model as hypothesized by the authors.

We agree that both of these experiments would be interesting and informative. Unfortunately, the electrophysiologist who was trained to perform these experiments is no longer in the lab. Thus, we plan to follow up with these experiments in a future study. This future direction is now included in the revised Discussion (page 23, line 518-520).

– It is not clear why a red-shifted channel rhodopsin (CrimsonR-tdTomato) was used as an anterograde tracer. Since optogenetics was not used in these experiments, control AAV that expressed only tdTomato would be appropriate. The rationale is not clear.

The anatomical experiments included in this manuscript were initially completed using a control AAV that expresses the red fluorophore mCherry. We noticed robust cytosolic expression in the CeA and terminal labeling throughout the brain with this approach but were concerned that the protein was not adequately reaching and labeling terminals. For this reason, we repeated the experiments using an AAV expressing ChrimsonR-tdTomato, which has been previously shown using electrophysiological approaches to efficiently (and functionally) reach terminals (10.1038/nmeth.2836;
10.1038/s41467-022-28539-7). Both of these AAV approaches are commonly used for neuroanatomical experiments in the field (10.1016/j.cub.2021.12.020; 10.1523/jneurosci.1515-19.2019; 10.1016/j.neuron.2019.03.037) and yielded comparable results in our experiments (summarized in Table 1), increasing our confidence in our findings. This rationale is included in page 5-6, line 109-130 of the revised manuscript.

– It is also difficult to appreciate CrimsonR-tdTomato transduced cells (high magnification images in Figure 1B).

Thank you for bringing this to our attention. We agree that the membrane expression of ChrimsonR-tdTomato makes it difficult to appreciate transduced cells in the original images (Sup Figure 1A-B in revised version). Given that transduction with mCherry is cytosolic and labels transduced cells, we have substituted the image in Figure 1 for an image of a brain transduced with mCherry. We preserved the ChrimsonR-tdTomato representative images in the Supplementary file for transparency purposes. As mentioned above (and summarized in Table 1 of the revised manuscript), the results of analyses of terminal expression using both ChrimsonRtdTomato and mCherry was similar. Moreover, as illustrated in Figure 1—figure supplement 1D, transduction efficiency was robust and comparable with both approaches.

– "ChrimsonR" should be changed to "ChrimsonR-tdTomato" throughout, because tdTomato is the fluorophore we see in the images.

"ChrimsonR-tdTomato" has been consistently used throughout.

– Figure 1- Supplemental figure 1(C) could benefit from higher magnification/resolution images of PKCδ terminals in the ZI. It looks like some terminals were labeled outside of ZI. Quantification of terminals in the ZI would be important because the focus of this study is on the CeA-PKCδ to ZI projections.

Thank you for bringing up these important points. We included higher resolution images of CeA-PKCδ terminals in the ZI at different rostral-caudal levels in Figure 1—figure supplement 2B. We also agree that a more detailed description of the terminal distribution and densities in the ZI is warranted. Thus, we performed additional experiments to thoroughly evaluate the anatomical distribution and densities of CeA-PKCδ terminals within the ZI of PKCδ-cre mice injected with AAV encoding ChrimsonR-tdTomato or mCherry in CeA. Consistent with our initial observations, our results show moderate to sparse labeling of CeA-PKCδ terminals that are dependent on the rostral-caudal level of the ZI. In the revised manuscript, we also demonstrate that these terminals are restricted to specific subregions within the ZI. Mapping of the ZI regions where we observed moderate terminal labeling in all injected brains are now included in Figure 1—figure supplement 2C. As requested by the reviewer, we quantified the densities of terminals in ZI regions as a function of the rostral-caudal level (Figure 1—figure supplement 2D). The results for these experiments are presented in page 8 and 9, line 178-194 and Figure 1—figure supplement 2 of the revised manuscript.

– Controls are needed for electrophysiology experiments to show blockade of optogenetically evoked IPSCs with bicuculine and a negative control at -70mV.

Thank you for this suggestion. We have completed the suggested additional electrophysiological experiments. The results are included in page 11, line 231-240 and Figure 2E and 2F of the revised version of the manuscript.

– Behavioral results are convincing but would benefit from more rigorous presentation. It is pretty much standard now to show individual data points to appreciate variability of the data. SEMs of behavioral data are amazingly small.

We agree and have included individual data points in all graphs in the revised version.

– Since modality specific behavioral effects were found, it would also be important to include other behavioral outcome measures (e.g., non-evoked).

We agree that evaluating the contribution of this circuit to non-evoked pain-related behaviors will be informative. However, given that those types of experiments are complex and would take many months to complete, we feel that it is beyond the scope of this paper. We have updated our Discussion in page 22, line 484-486 to acknowledge that this is an important future direction.

– Consistent use of terminology for the viral vectors is suggested.

Done